# On the Measurements of the Surface-Enhanced Raman Scattering Spectrum: Effective Enhancement Factor, Optical Configuration, Spectral Distortion, and Baseline Variation

**DOI:** 10.3390/nano13232998

**Published:** 2023-11-22

**Authors:** Yiping Zhao

**Affiliations:** Department of Physics and Astronomy, The University of Georgia, Athens, GA 30602, USA; zhaoy@uga.edu; Tel.: +1-706-542-7792

**Keywords:** surface-enhanced Raman scattering, enhancement factor, optical attenuation, spectral distortion, baseline, effective medium theory

## Abstract

In this paper, a comprehensive theoretical framework for understanding surface-enhanced Raman scattering (SERS) measurements in both solution and thin-film setups, focusing on electromagnetic enhancement principles, was presented. Two prevalent types of SERS substrates found in the literature were investigated: plasmonic colloidal particles, including spherical and spheroid nanoparticles, nanoparticle diameters, and thin-film-based SERS substrates, like ultra-thin substrates, bundled nanorods, plasmonic thin films, and porous thin films. The investigation explored the impact of analyte adsorption, orientation, and the polarization of the excitation laser on effective SERS enhancement factors. Notably, it considered the impact of analyte size on the SERS spectrum by examining scenarios where the analyte was significantly smaller or larger than the hot spot dimensions. The analysis also incorporated optical attenuations arising from the optical properties of the analyte and the SERS substrates. The findings provide possible explanations for many observations made in SERS measurements, such as variations in relative peak intensities during SERS assessments, reductions in SERS intensity at high analyte concentrations, and the occurrence of significant baseline fluctuations. This study offers valuable guidance for optimizing SERS substrate design, enhancing SERS measurements, and improving the quantification of SERS detection.

## 1. Introduction

Surface-enhanced Raman scattering (SERS) is a powerful spectroscopy technique that has extensively been employed for chemical and biological sensing. When target analytes are in close proximity to specially designed nanostructured surfaces (or plasmonic nanostructures), the Raman signal of the target analytes can be significantly enhanced due to local electromagnetic field enhancement and possible chemical enhancement due to charge transfer [1,2]. With enhancement factors typically ranging from 10^6^ to 10^8^, SERS exhibits remarkable sensitivity, capable of detecting molecules at exceptionally low concentrations, sometimes even at the single-molecule level [3]. The intrinsic vibrational modes of analytes impart distinct patterns to SERS spectra and can be treated as molecular fingerprints. This characteristic grants SERS spectra high selectivity (or specificity), enabling the identification of specific molecules within complex metrices. This specificity forms the foundation for SERS to be considered a label-free detection method, and SERS has found widespread applications in the detection and identification of a diverse array of chemical and biological analytes. Its applications span various domains within the chemical and biological sensor community, encompassing areas such as medical diagnostics, drug discovery, food safety, and environmental monitoring, among others [4].

Many interpretations of SERS results in the existing literature are rooted in a number of implicit assumptions, specifically that SERS hot spots, where the most intense local electric fields exist, predominantly influence SERS spectrum generation in addition to chemical enhancement. Although it has been widely acknowledged that electromagnetic enhancement indeed plays a significant role in determining SERS spectra, practical SERS spectrum measurements often reveal other phenomena that cannot be solely explained by the SERS enhancement factor (EF) or hot spots. From our own experience, we noticed that spectral features of SERS spectra from analytes with the same SERS substrates can vary when measured from one location to another. Additionally, SERS spectra usually display significant fluctuations in baseline from one location to another. The SERS EF has primarily been defined using Raman reporter molecules and has rarely been discussed in the context of detecting large analyte particles. Hence, there is a compelling need for a thorough investigation into the intricate details of SERS measurements to comprehend how various parameters could contribute to SERS measurements effectively.

Upon a more detailed analysis of SERS-based measurements, it has become apparent that a multitude of intricate physical and chemical processes are potentially in play. Most of all, the SERS measurement configuration, the SERS substrate, and the target analyte play dominant roles in determining the final measurement result. In terms of measurement configuration, SERS measurements can broadly be categorized into solution-based detection and film-based detection. In each measurement configuration, there will be different types of SERS substrates which exhibit different physical and chemical properties. Finally, whether the analyte’s size can accommodate the dimension of the hot spot determines what kind of ideal SERS EF a system can achieve.

In solution-based measurements, plasmonic colloidal particles (PCNs) are uniformly dispersed in the analyte solution. Analytes adhere to the PCNs, and upon exposure to the appropriate excitation laser, SERS signals can be directly obtained from this PCN suspension. In this measurement setup, several processes can significantly influence the final SERS spectrum: (1) the analyte adsorption process, including the quantity of analyte adsorbed on the PCNs, the adsorption location (whether it is in a hot spot), and the orientation of the adsorbed analytes; (2) the polarization of the excitation laser, which can influence the hot spot locations; and (3) the optical path during Raman excitation and signal collection. The PCN suspension can be treated as an optical medium composed of the PCNs and the analytes. Challenges emerge as the excitation laser must be precisely focused within the suspension, potentially causing laser intensity attenuation within the medium. Furthermore, the scattered signal must propagate through the medium for signal collection, a process that can also be optically modulated via the medium itself. Any variation in analyte concentration or fluctuation in PCN concentration may alter the optical properties of this medium. Concurrently, chemisorption and physisorption take place between the PCNs and analyte molecules, further modifying the medium’s overall optical properties.

On the other hand, thin-film-based measurements involve the applying the analyte solution, either drop-cast onto the substrate or with the substrate immersed in the solution. The sample preparation inherently involves equilibrium or non-equilibrium wetting/dewetting processes. In the meantime, since the SERS active layer must be supported with a substrate, multiple interfaces are encountered by both the excitation laser and the collected SERS signal during the measurement. Additionally, the intrinsic optical properties of the SERS active layer, other supporting layers, as well as the analyte can play a pivotal role. Whether the analyte significantly absorbs within the wavenumber region of the SERS spectrum or produces a fluorescence signal significantly influences the spectrum’s shape. These intricate considerations underline the complexity inherent in SERS measurements.

This study thoroughly examined the processes mentioned above and the associated parameters that impact the determination of an effective SERS EF from a theoretical perspective, especially the change in the spectral shape, the modification in SERS quantification, as well as the variation in the SERS baseline. General mathematical equations were provided to directly link SERS intensity with its relevant parameters. It is important to note that these discussions were based on the assumption that only the local electromagnetic enhancement, specifically the hot spot, plays the dominant role in these phenomena.

## 2. Overview of the SERS Signal

The SERS signal in any measurement can be generally expressed as:(1)$$I_{SERS}\left(\Delta v\right)=R_{in}\left(\Delta v\right)\left(I_{AH}+I_{AR}+I_{BH}+I_{BR}+I_{MH}+I_{MR}+I_{BS}+I_{FLU}+I_{bk}\right)+I_{noise},$$
where $R_{in}\left(\Delta v\right)$ is the instrument response function, encompassing the quantum efficiency of the detector and the spectral response of each optical component in the instrument. $I_{AH}$, $I_{BH}$, and $I_{MH}$ denote the SERS intensity originating from the analyte, background, and medium molecules adsorbed on SERS hot spots, respectively, often dominating the spectrum. Correspondingly, $I_{AR}$, $I_{BR}$, and $I_{MR}$ represent the Raman signals of these molecules in non-hot spot locations. $I_{BS}\left(\Delta v\right)$ accounts for potential fluorescent signals from the analyte, background, or other non-target molecules in the specimen and solvent, or any non-Raman contributions from the SERS structures that give rise to the baseline of the spectrum. $I_{FLU}$ signifies fluctuating SERS (or Raman signal) due to sampling or other measurement configurations. $I_{bk}$ denotes the background signal resulting from illumination, which is eliminable in instrument design. Finally, $I_{noise}$ represents the electronic noise inherent to the Raman instrument, independent of the instrument’s optical response (except for the detector). Both $I_{BH}$ and $I_{MH}$ represent interference SERS spectra, which can significantly impact SERS spectral analysis. The SERS intensity $I_{iH}$ (i=A, B, and M) from analytes in SERS hot spots can be written as:(2)$$I_{iH}=G_{SERS}^{0}F_{iH}\sigma_{iH}n_{iH}N_{H}I_{0},$$
where $G_{SERS}^{0}$ represents the theoretical SERS EF at the hot spot location and remains constant regardless of the types of analytes, provided they are significantly smaller than the hot spot dimensions. Theoretically, $G_{SERS}^{0}$ should be influenced by the specific adsorption locations of analytes on the SERS substrates due to the varying local electric field (E-field) at different substrate points. However, for simplification purposes, it is often treated as a constant (or sometimes derived from the average electromagnetic enhancement across the entire substrate area, as observed in several studies [5]). $F_{iH}$ denotes the fraction of photons emitted by analytes within a hot spot and collected via the microscopic objective. $\sigma_{iH}\left(\Delta v\right)$ denotes the SERS cross-section of corresponding analytes at a specific wavenumber, Δv. $n_{iH}$ stands for the number of analytes adsorbed in a SERS hot spot, while $N_{H}$ is the total number of hot spots in the measurement volume, assuming equal contribution from each hot spot. $I_{0}=I_{0}\left(\lambda_{ex}\right)$ indicates the incident intensity of the excitation laser at a wavelength of $\lambda_{ex}$. The normal Raman intensity, $I_{iR}$, can be expressed as:(3)$$I_{iR}=F_{iR}N_{iR}\sigma_{iR}I_{0},$$
with a collected fraction, $F_{iR}$, of photons, the total number, $N_{iR}$, and the Raman scattering cross-section,$\;\sigma_{iR}$, of corresponding Raman scatterers. $I_{FLU}$ can be written as:(4)$$I_{FLU}=\underset{i}{\sum }\left(\Delta I_{iH}+\Delta I_{iR}\right),$$
where
(5)$$\Delta I_{iH}=G_{SERS}^{0}F_{iH}\sigma_{iH}N_{H}I_{0}\Delta n_{iH}+G_{SERS}^{0}F_{iH}\sigma_{iH}n_{iH}I_{0}\Delta N_{H},$$
(6)$$\Delta I_{iR}=F_{iR}\sigma_{iR}I_{0}\Delta N_{iR},$$
and $\Delta n_{iH},\;\Delta N_{H}$, and $\Delta N_{iR}$ represent fluctuations in $n_{iH}$, $N_{H}$, and $N_{iR}$ during the SERS measurement, respectively. It was assumed that there was no fluctuation in $I_{0}$.

Clearly, the nine contributions, $I_{iH}\left(\times 3\right),I_{iR}\left(\times 3\right),I_{FL},I_{FLU},\mathrm{and}\;I_{bk}$, are channeled through the optics of the instrument. Consequently, the resultant SERS spectrum acquired via the Raman instrument is contingent upon the magnitude of each intensity, which is influenced by several factors. If the SERS signal predominates the total intensity,$\;I_{total}\left(\Delta v\right)$, the spectrum (both intensity and spectral shape) will be influenced by the following factors: (1)Instrument characteristics, including the spectral response of the instrument.(2)Excitation laser parameters, such as its wavelength, incident angle, and polarization.(3)Signal collection setup comprising scattering angle and collection solid angle.(4)SERS substrate properties encompassing size, shape, topology/morphology of the active SERS structure, uniformity, contamination, and dynamic effects.(5)Analyte properties involving the size of the analytes, intrinsic Raman scattering cross-section, potential fluorescence signal, optical response, and more.(6)Analyte adsorption characteristics, such as adsorption affinity, distance to the SERS substrate, orientation, whether it involves equilibrium or non-equilibrium adsorption, or competing adsorption for multiple analytes.(7)Surface modifications/contamination: on the SERS substrate or within the medium where the analyte is dissolved, if the SERS substrate is modified or functionalized via specific cap agents, or if contaminants are acquired by the SERS substrate in air or in solution, or due to storage, or if the SERS analyte is dissolved in a medium containing other analytes, these additional analytes may adsorb on hot spot locations, generating additional SERS signals, e.g., $I_{BH}$ and $I_{MH}$.

As shown in Equation (2), $I_{iH}$ is fundamentally determined by six parameters, namely $G_{SERS}^{0}$, $F_{iH}$, $\sigma_{iH}$, $n_{iH}$, $N_{H},$ and $I_{0}$. $F_{iH}$ depends on the instrument design, the output laser intensity, and the specific SERS substrate properties. Once the instrument design, laser intensity, and substrate characteristics are established, we can treat $F_{iH}=F_{H}$, i.e., $F_{iH}$ is a constant. The value of $\sigma_{iH}$ relies on the intrinsic properties of the SERS scatterers, the SERS substrate, the affinity between the SERS scatterers and the substrates, as well as the polarization of the excitation light. Both $F_{H}$ and $\sigma_{iH}$ are set once the measurement system and the analyte/SERS substrate system are defined. The remaining four parameters, $G_{SERS}^{0}$, $n_{iH}$, $N_{H}$, and $I_{0}$, emerge as the most crucial factors in determining $I_{SERS}$. Both $G_{SERS}^{0}$ and $n_{iH}$ are interrelated and influenced by various experimental conditions, such as the configuration of the SERS substrates and the adsorption kinetics of the analyte, among others. $N_{H}$ is determined by the design and engineering of the SERS substrate, alongside the accessibility for analytes. Meanwhile, the actual $I_{0}$ experiences attenuation due to the optical path taken by the excitation laser beam and the backscattered SERS signal.

Practically, both $G_{SERS}^{0}$ and $n_{iH}$ cannot be directly determined through experimentation. Instead, most researchers employ the apparent EF or effective EF, denoted as $G_{SERS}^{e}$, to account for the SERS EF of a particular analyte:(7)$$G_{SERS}^{e}=\frac{I_{SERS}/N_{A}}{I_{Raman}/N_{R}},$$
where $N_{A}$ is the total number of the analytes probed by the excitation laser, and $I_{Raman}$ represents the Raman signal from a bulk volume solution of the same analyte, with the total number of the scattering analytes to be $N_{R}$. To make $G_{SERS}^{e}=G_{SERS}^{0}$, according to Equations (2) and (3), at least five assumptions need to be made in Equation (7): (1) the other seven contributions in Equation (1), namely $I_{BH},\;I_{BR},I_{MH},I_{MR},I_{FL},I_{FLU},\;\mathrm{and}\;I_{bk}$, are negligible; (2) the instrument’s collection efficiencies, $F_{H}$ and $F_{R}$, shall be the same; (3) the incident excitation laser intensities are the same; (4) $\sigma_{iH}=\sigma_{iR}$; and (5) $N_{A}=n_{AH}N_{H}$, i.e., all the probed analytes under the excitation laser beam are located in the hot spots. While the first three assumptions may be valid or deliberately designed to be valid, $n_{AH}N_{H}$ typically represents only a small fraction of $N_{A}$ in most measurement configurations, depending on the sizes of the hot spots and the analytes. Therefore, in general, $G_{SERS}^{e}$ should be significantly smaller than $G_{SERS}^{0}$. In reality, even though the definition in Equation (7) is relatively straightforward experimentally, it encapsulates multiple hidden factors, as highlighted by Le Ru et al. [6]. Based on Equations (2) and (7), Equation (1) can be redefined as follows:(8)$$I_{SERS}=R_{in}I_{AH}=G_{SERS}^{e}R_{in}F_{H}N_{A}\sigma_{AH}I_{0}.$$
Note that here $\sigma_{AH}$ will also be an effective SERS cross-section and:(9)$$I_{Raman}=R_{in}F_{R}N_{R}\sigma_{AR}I_{0}.$$

However, an often overlooked assumption in the existing literature pertains to the alteration in the optical response of the measurement system when obtaining $I_{AH}$ and $I_{AR}$. $I_{AH}$ is measured when the target analytes adsorb onto the SERS substrate, while $I_{AR}$ is obtained either from a high-concentration solution or powder of the analyte. Thus, the optical behaviors of the targeted system in these two measurements can diverge significantly. Moreover, there are typically two distinct types of SERS measurements: one involves a solution with suspended nanoparticle-based SERS substrates, and the other utilizes thin-film-based SERS substrates. Different SERS substrates can introduce varied optical responses into $I_{total}$, implying that both Equations (8) and (9) need to be adjusted:(10)$$I_{SERS}=G_{SERS}^{e}R_{SERS}R_{in}F_{H}N_{A}\sigma_{AH}I_{0},$$
(11)$$I_{Raman}=R_{R}R_{in}F_{R}N_{R}\sigma_{AR}I_{0},$$
where $R_{SERS}$ and $R_{R}$ denote the optical responses in SERS and Raman measurements, respectively. Based on Equation (7), the experimentally observed SERS EF $G_{m}$ can be formulated as:(12)$$G_{m}=\frac{I_{SERS}/N_{A}}{I_{Raman}/N_{R}}=\frac{R_{SERS}}{R_{R}}G_{SERS}^{e}.$$

Thus, if the SERS measurement configuration exhibits a strong optical response from the SERS substrate–analyte system, this response will significantly impact the determination of the SERS EF and other spectroscopic relationships. In fact, most SERS substrates are designed to showcase a strong optical response. For example, from Van Duyne’s work, the excitation wavelength, $\lambda_{ex}$, for plasmonic SERS substrates will be chosen to be close to its localized surface plasmon resonance (LSPR) wavelength, $\lambda_{LSPR}$, to achieve high SERS enhancement [7]. Around the $\lambda_{LSPR}$, the substrate is extremely absorptive. In the following discussion, we will explore the effects of Equations (10) and (11) on the determination of $G_{SERS}^{e}$, $G_{m}$, and other SERS spectral characteristics for different SERS measurement configurations.

## 3. The Measured SERS Enhancement Factor $G_{m}$

### 3.1. Solution-Based SERS Measurements

The solution-based SERS measurement setup is depicted in Figure 1. SERS nanoparticles (PCNs) are uniformly suspended in a solution, with analytes evenly adsorbed on the PCN surfaces. The excitation laser is focused at a distance of *f* within the suspension. SERS signals are collected using a backscattering configuration, specifically from the PCNs within a liquid volume outlined by the dashed blue square in the figure. In order to derive the final expression of SERS intensity, we need to consider two scenarios: firstly, when the analyte molecules are significantly smaller than the size of the hot spots in PCNs, which constitutes the majority of situations in SERS measurements; and secondly, when the analytes are much larger than the size of PCNs. This latter case can occur when the target analytes are viruses, bacteria, or even tissues.

#### 3.1.1. Analytes Much Smaller than the Size of the Hot Spots

In the plasmonic research community, it has been well established that the hot spot size of a PCN is typically in the range of 5–10 nm near its surface. When the size of the analytes is much smaller than the hot spot size, these analytes can adsorb onto hot spot locations, generating substantial SERS signals. Given that an effective EF is in the range from 10^6^ to 10^8^, even a small fraction of analytes adsorbed inside the hot spots can dominate the collected Raman signal. Consequently, understanding the factors influencing $G_{SERS}^{e}$ during the SERS measurement is crucial.

According to Le Ru et al. [6], various factors can impact $G_{SERS}^{e}$, including:(1)The excitation wavelength, $\lambda_{ex}$.(2)Polarization of the excitation laser.(3)PCN morphology.(4)Variation in PCN size and shape.(5)Orientation of the adsorbed analytes.(6)Fraction of analytes in hot spot locations.

Firstly, regarding the average EF $G_{SERS}^{A}$ for a single PCN, it is important to note that the discussions presented here focus on scenarios involving sub-monolayer or single monolayer coverage of analytes on a PCN.

***Spherical PCNs*:** In solution-based detection, the behavior of dispersed PCNs largely influences $G_{SERS}^{A}$, determined by the shape, size, and aggregates of these PCNs. Consider a scenario where PCNs are individual Au or Ag nanoparticles with a specific $\lambda_{LSPR}$. When $\lambda_{ex}$ is very close to $\lambda_{LSPR}$, the SERS signal is maximized [7]. Let us assume all PCNs are spherical in shape (Figure 2A). The estimation of the average $G_{sphere}^{A}$ depends on the following factors: the polarization of the excitation laser, the orientation of the adsorbed analytes, analyte coverage, and PCN Brownian motion.

In the case of a vertically linear polarized excitation laser, hot spots on a spherical PCN are typically located at the top and bottom poles of the PCN, aligned with the polarization direction (Figure 2A). If an analyte adheres to the top surface of the PCN, with its long axis perpendicular to the surface, the Raman active mode ($\Delta v_{∥}$ mode) with vibrational components along the analyte’s axis will be enhanced. However, if the analyte’s orientation on the PCN surface rotates by 90 degrees, as depicted on the bottom surface in Figure 2A, the $\Delta v_{∥}$ mode will not be enhanced. Instead, the Raman active mode with a vibrational component perpendicular to the molecule’s axis ($\Delta v_{⊥}$ mode) will be enhanced. This non-uniform enhancement of vibrational modes can alter the shape of the SERS spectrum.

Representing the SERS scattering cross-sections of the analyte with its axis parallel ($\Delta v_{∥}$) and perpendicular ($\Delta v_{⊥}$) to the polarization direction as $\sigma_{AH}^{∥}$ and $\sigma_{AH}^{⊥}$, respectively, and considering the orientation distribution of analyte molecules as $P_{O}\left(\theta ,\varphi \right)$ (refer to Figure 2C) with respect to the polarization direction, the SERS EF $G_{sphere}^{D}$, accounting for the orientation effect, can be expressed as follows:(13)$$G_{sphere}^{D}=\frac{G_{sphere}^{0}}{{\bar{\sigma }}_{AH}^{D}}\int_{0}^{2\pi }d\varphi \int_{0}^{\pi }\left(\sigma_{AH}^{∥}\cos^{2}\theta +\sigma_{AH}^{⊥}\sin^{2}\theta \right)P_{O}\left(\theta ,\varphi \right)\sin\theta d\theta ,$$
where $G_{sphere}^{0}$ is the ideal EF of a spherical PCN when an analyte adsorbs on the hot spot, $\int_{0}^{2\pi }d\varphi \int_{0}^{\pi }P_{O}\left(\theta ,\varphi \right)\sin\theta d\theta =1$, and ${\bar{\sigma }}_{AH}^{D}$ is the average SERS scattering cross-section at Δv:(14)$${\bar{\sigma }}_{AH}^{D}\left(\Delta v\right)=\frac{\sigma_{AH}^{∥}+\sigma_{AH}^{⊥}}{2}.$$

Consider the comparison between a scenario where analyte molecules are randomly adsorbed (Figure 2C) and a case where analyte molecules are well oriented due to self-assembly (Figure 2D). In Figure 2D, the SERS spectrum is primarily governed by $\sigma_{AH}^{∥}\left(\Delta v\right)$, whereas in Figure 2C, both $\sigma_{AH}^{∥}\left(\Delta v\right)$ and $\sigma_{AH}^{⊥}\left(\Delta v\right)$ contribute to the final SERS spectrum. It is evident that if the analyte possesses a complex structure with varying symmetry, Equation (13) would become more intricate. Consequently, due to potential changes in analyte orientation, not only can the shape of the SERS spectrum be altered but also the effective EF may differ at various Δv values.

In solution-based SERS measurements, PCNs undergo Brownian motion both translationally and rotationally. Therefore, $G_{sphere}^{A}$ represents the average of $G_{sphere}^{D}\left(\Delta v\right)$ when the sites of adsorbed analytes become hot spot locations. Assuming a very low analyte density (as depicted in Figure 2C), where only a few analytes (*M_A_*) are adsorbed on the PCN surface, let us consider that the hot spot has a solid angle of $\Omega_{H}$ in the spherical PCN, denoted as $\Omega_{H}=2\pi \left[1-\sqrt{1-h^{2}/r^{2}}\right]$, where *h* is the projected radius of the hot spot on the spherical PCN, and *r* is the PCN radius. When the probability of an analyte in a hot spot location is $2\times M_{A}\frac{\Omega_{H}}{4\pi }$, the average $G_{SERS}^{A}$ for a single PCN becomes:(15)$$G_{sphere}^{A}=G_{sphere}^{D}\frac{\Omega_{H}}{2\pi }.$$

If a PCN is entirely coated with a layer of analytes, these analytes may tend to align around the PCNs in a specific orientation, as illustrated in Figure 2D. In this case, irrespective of the PCN’s orientation, there will always be analyte molecules present in the hot spot locations. Let us assume that each analyte occupies a small solid angle, $\Omega_{A}$, on the surface of a PCN. Given that there are always $2\times \frac{\Omega_{H}}{\Omega_{A}}$ analytes situated in a hot spot, the average $G_{sphere}^{A}$ for a single PCN becomes:(16)$$G_{sphere}^{A}=G_{sphere}^{D}\frac{2\Omega_{H}}{\Omega_{A}M_{A}}.$$

Here, $M_{A}=4\pi /\Omega_{A}$, i.e., the equation $G_{sphere}^{A}=G_{sphere}^{D}\Omega_{H}/2\pi$ holds, making Equation (16) equivalent to Equation (15). However, Equation (16) remains constant over time, whereas Equation (15) represents a time-averaged result, depending strongly on random motion. This dependence could offer a method to measure PCN size, similar to the principles employed in dynamic light scattering [8].

When unpolarized light is used for excitation, hot spots will form around the equatorial band of the PCN, as depicted in Figure 2B. This is due to the electric fields being equally distributed in all directions perpendicular to the light’s incident direction. Although this change in polarization does not significantly impact the distribution of analyte orientations in the final SERS spectrum (i.e., the discussion of $S_{∥}\left(\Delta v\right)$ and $S_{⊥}\left(\Delta v\right)$ for Equation (13) remains valid), the projected intensity of the excitation laser in a specific direction reduces to $I_{0}/2$. As shown in Figure 3, taking into account the probability of analytes being adsorbed in the hot spot area $\frac{2\pi rh}{4\pi r^{2}}=\frac{h}{2r}$, we obtain:(17)$$G_{sphere}^{A}=G_{sphere}^{D}\frac{h}{4r}.$$
When h/r≪1, $\Omega_{H}\approx \pi h^{2}/r^{2}$, making Equations (15) and (16) to become $G_{sphere}^{A}=G_{sphere}^{D}h^{2}/r^{2}$, which is smaller than the $G_{sphere}^{A}$ obtained in Equation (17). Therefore, in the context of spherical PCN suspension in solutions, using unpolarized excitation light can yield a higher $G_{sphere}^{A}$.

**Figure 3 nanomaterials-13-02998-f003:**
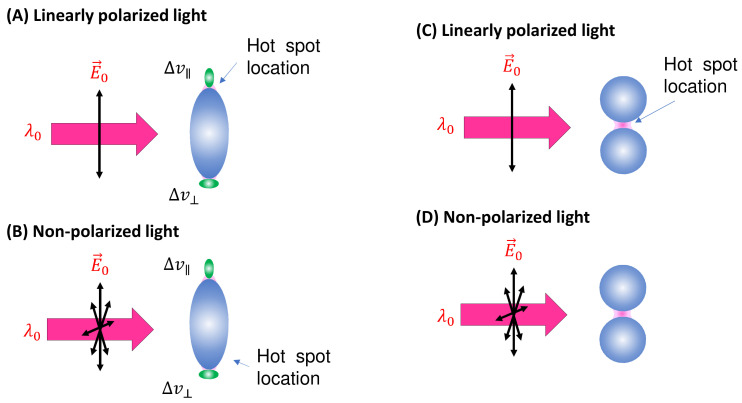
(**A**) The linearly polarized and (**B**) non-polarized excitation and possible analyte orientations on a spheroid PCN. (**C**) The linearly polarized and (**D**) non-polarized excitation on a PCN dimer.

In addition, experimentally, there is always a distribution of the size, *s*, and shape, Σ, of the PCNs, or even an aggregation of PCNs. In this case, $\lambda_{LSPR}$ is a function of s, Σ, and aggregations, and $G_{sphere}^{0}$ is not a constant, and neither is $G_{sphere}^{A}$. Thus, the effective $G_{sphere}^{e}$ at a particular $\lambda_{ex}$ can be expressed as:(18)$$G_{sphere}^{e}=\iint_{s,\Sigma }^{}G_{sphere}^{A}\left(s,\Sigma \right)P\left(s,\;\Sigma \right)dsd\Sigma ,$$
where $P\left(s,\;\Sigma \right)$ is the probability density function of *s* and ∑, with $\iint_{s,\Sigma }^{}P\left(s,\;\Sigma \right)dsd\Sigma =1$.

Certainly, if two or more PCNs aggregate, as outlined by the red dashed ovals in Figure 1, the $\lambda_{LSPR}$ can undergo a significant red shift due to plasmonic coupling/hybridization [9,10]. Therefore the contribution from aggregate particles to the final SERS intensity can often be neglected. However, if $\lambda_{ex}$ is tuned to the $\lambda_{LSPR}$ of the aggregated PCNs, the primary contribution to SERS will stem from the aggregated PCNs, not the monodispersed ones.

***Spheroid PCNs*:** If the PCNs are anisotropic, like the spheroid particles shown in Figure 3A, the estimation of $G_{SERS}^{e}$ will be extremely different. Monodispersed PCN spheroids possess two LSPR wavelengths (specifically considering prolate PCNs): a longitudinal mode ($\lambda_{LSPR}^{L}$) excited along the axis of the spheroid, and a transverse mode ($\lambda_{LSPR}^{T}$) with resonance direction perpendicular to the spheroid’s axis [11]. Depending on the aspect ratio of the spheroid, the values of $\lambda_{LSPR}^{L}$ and $\lambda_{LSPR}^{T}$ could be very close (in the case of a small aspect ratio) or significantly apart from each other (in the case of a high aspect ratio). When linearly polarized light with $\lambda_{ex}$ ≈ $\lambda_{LSPR}^{L}$ excites the PCN spheroid along its axis, a very high local electric field ($E_{L}^{L}$) appears at the two poles along the axis. On the other hand, when linearly polarized light with $\lambda_{ex}$ ≈ $\lambda_{LSPR}^{T}$ is employed perpendicular to its axis, the local electric fields ($E_{L}^{T}$) at the two poles (the hot spot locations) perpendicular to the axis exhibit a much smaller magnitude than $E_{L}^{L}$. Typically, researchers opt to use $\lambda_{ex}$ ≈ $\lambda_{LSPR}^{L}$ to generate SERS signals from PCN spheroids. In this case, unlike the situation with spherical PCNs, the hot spots are site-specific. Specifically, SERS signals are only produced when analyte molecules are adsorbed on the two poles of the spheroid, given that the spheroid’s long axis partially aligns with the polarization direction. Thus, the average EF $G_{spheriod}^{A}$ for a single spheroid is influenced by the orientation of analytes in the hot spots, the likelihood of analytes being inside the hot spots, and the orientation of the spheroid with respect to the polarization direction.

To explore the effect of analyte molecule adsorption orientation, Equation (13) is valid for this context. To estimate the probability of analytes inside the hot spot, we considered two scenarios: analytes having an equal likelihood of adsorbing on any surface location of the PCN, and the adsorption probability depending on the curvature of the location [12,13].

Considering the first scenario, we can maintain the assumption that the hot spot on the tip of the spheroid projects a circular area with a radius of *h* on the spheroid. Assuming that the long axis radius of the spheroid is *c* and the short axis radius is *a*, the probability of finding one of the $M_{A}$ total adsorbed analytes located at the two hot spots is outlined by:(19)$$P_{prolate}^{L}=2\times \frac{c\left\{\frac{\arcsin\left(e\right)-\arcsin\left(e_{1}\right)}{e}+\frac{a}{c}-\left(1-\frac{b}{c}\right)\sqrt{1-e_{1}^{2}}\right\}}{2a\left(1+\frac{c}{ae}\right)},$$
where the numerator in Equation (19) is the area of the hot spot on a pole of the spheroid, and the denominator is the total area of a PCN, i.e., $e^{2}=1-a^{2}/c^{2}$, $e_{1}=e\left(1-b/c\right)$, and $b=c\left(1-\sqrt{1-h^{2}/a^{2}}\right)$. Thus, the average $G_{spheriod}^{A}$ for a single prolate PCN can be written as:(20)$$G_{spheriod}^{A}=G_{spheriod}^{LD}P_{prolate}^{L}.$$

However, since only the external electric field parallel to the axis of the spheroid can generate the SERS signal, if these two vectors make an angle (θ), then the contribution of this particularly orientated spheroid to the SERS signal can be written as:(21)$$dI_{SERS}^{spheriod}\left(\theta \right)\propto G_{spheriod}^{LD}P_{prolate}^{L}\cos^{2}\theta \sin\theta d\theta d\varphi .$$

Considering that the orientation of the spheroid particle can be uniformly distributed at any orientation due to Brownian motion, we eventually obtain:(22)$$G_{spheriod}^{A}=\frac{1}{3}G_{spheriod}^{LD}P_{prolate}^{L}.$$

For the second scenario, if the analyte’s adsorption probability depends on the local curvature of a PCN, then Equation (19) can be rewritten as:(23)$$P_{prolate}^{L}=\iint_{Hot\;spot}^{}p\left(\gamma \right)dA,$$
where $p\left(\gamma \right)$ is the curvature (γ) dependent adsorption probability density of analytes on a small surface, *dA*. The integration is conducted over the entire hot spot area. Except for the calculation of $P_{prolate}^{L}$, the other expressions for the EF remain the same. Equation (22) denotes the result of orientational averaging of the PCN spheroids. Clearly, if all the spheroid particles could be aligned along the polarization direction, the maximum SERS signal could be obtained from the spheroid PCNs.

For non-polarized excitation, only the light polarized along the long axis of the spheroid can excite the SERS signal, accounting for only $I_{0}/2$. Assuming that analyte adsorption is independent of the curvature, then:(24)$$I_{SERS}^{spheriod}=G_{spheriod}^{LD}P_{prolate}^{L}F_{AH}N_{A}{\bar{\sigma }}_{AH}^{D}\frac{1}{2}I_{0},$$
i.e.,
(25)$$G_{spheriod}^{A}=\frac{1}{2}G_{spheriod}^{LD}P_{prolate}^{L}.$$

Thus, compared to Equation (22), the $G_{spheriod}^{A}$ for non-polarized excitation (Equation (25)) is larger than that of linear polarization.

***Spherical PCN dimers*:** Another typical PCN configuration is a spherical colloid dimer with extremely small gaps, ranging from 1 to 5 nm, as depicted in Figure 3C [14]. Clearly, such a dimer particle is also anisotropic, meaning that the formation of hot spots depends on the polarization of the incident light. Moreover, to obtain a high SERS intensity, the analytes must be located within the gaps; if the analytes are outside the gaps, the SERS signal will be significantly reduced.

The calculation of $G_{dimer}^{A}$ for PCN dimers is similar to that for PCN spheroids, as the hot spot is location-specific, and its excitation is highly dependent on the relative orientation of the dimer’s long axis and the polarization direction. Therefore, all the discussions applicable to spheroid PCNs are also valid for PCN dimers. As the dimer consists of two spheres, if there are no other effects, and the analytes have an equal probability of adsorbing on any surface location of the PCNs (considering only surface adsorption), then:(26)$$P_{dimer}=\frac{\Omega_{H}}{4\pi },$$
and the $G_{SERS}^{A}$ for a single PCN dimer is:(27)$$G_{dimer}^{A}=G_{dimer}^{D}P_{dimer}.$$

Considering the orientation distribution of the dimers in the solution, according to Equations (21) and (22), one has:(28)$$G_{dimer}^{A}=\frac{\pi }{4}G_{dimer}^{D}P_{dimer}.$$

For non-polarized light, the discussions for spheroid PCNs can be applied, and Equation (25) is valid.

Practically, both PCN spheroids and dimers exhibit size and shape distributions. Therefore, the derived Equations (22), (25), and (28) must undergo shape and size averaging, similar to the process outlined in Equation (18), to determine the ultimate $G_{dimer}^{e}$.

#### 3.1.2. Analytes Much Larger than the Size of the PCNs

If the analyte is not a small molecule but a biological organism, like a virus, bacteria, or tissues, the expression of SERS intensity in solution-based measurements diverges significantly from those in Section 3.1.1 as the adsorption configuration of PCNs and analyte particles is changed. The PCNs can only adsorb onto a very small fraction of the surface of the analyte particles, as shown in Figure 4, and the local electric field from the hot spot would penetrate into the analyte’s surface following either an exponential or power law decay relationship. In other words, molecules from various depths within the analyte’s surface would contribute to the overall SERS spectrum. Let $G_{AH}^{0}\left(z\right)=G^{0}e^{-\frac{4z}{\delta }}\;$($G\propto {\left|E_{loc}\right|}^{4}$) [15,16], and consider an ideal scenario with a spherical PCN, as shown in Figure 4; the layer density of molecules on the analyte’s surface, $\eta_{M}\left(z\right)$, varies with depth, leading to distinct SERS scattering cross-sections ($\sigma_{AH}\left(z\right)$). Thus, the effective SERS intensity, $I_{AH}^{S}$, from a single hot spot can be expressed as:(29)$$I_{AH}^{S}=F_{AH}I_{0}\frac{4}{\delta }\int_{0}^{\infty }\eta_{M}\left(z\right)\sigma_{AH}\left(z\right)G_{SERS}^{0}e^{-\frac{4z}{\delta }}\pi h^{2}dz.$$

In this case, defining a SERS EF becomes impractical for several reasons. Firstly, the depth-dependent nature of the analyte particle may not be uniform; different layers at various depths could contain diverse molecules, such as viruses or bacteria, each contributing distinct SERS spectral features. Secondly, accurately estimating the number of molecules contributing to the final SERS spectrum is exceedingly challenging. Finally, determining the contribution of molecules from the limited layer of the analyte particle to the normal Raman intensity presents a formidable task. Due to the high complexity and inhomogeneity of analyte particles, the Raman spectrum is not only influenced by the surface components of the particle but also by contents inside the particles. As a result, the SERS spectrum and the Raman spectrum may exhibit significant differences. Moreover, determining the number of specific molecules responsible for the Raman spectrum is exceptionally difficult. Nevertheless, if we assume that the analyte particle is uniform and possesses a constant surface density (such as a polystyrene colloidal bead), denoted by $\eta_{M}\left(z\right)=\eta_{M0}$, $\sigma_{AH}\left(z\right)={\bar{\sigma }}_{AH}$, then:(30)$$I_{AH}^{S}=F_{AH}I_{0}\eta_{M0}\pi h^{2}G_{SERS}^{0}{\bar{\sigma }}_{AH}=G_{SERS}^{0}F_{AH}{\bar{\sigma }}_{AH}N_{A}I_{0},$$
where the number of the analyte molecules contributing to SERS is $N_{A}=\eta_{M0}\pi h^{2}$.

***Spherical PCNs*:** When a linearly polarized excitation is applied, and the spherical PCNs are significantly smaller than the size of the analyte particle, they can randomly adsorb onto the analyte surface with equal probability. In this scenario, only PCNs adsorbed in locations with a local surface normal component aligned with the polarization direction can generate the SERS signal. This condition applies to PCN particles numbered one, two, six, and seven in Figure 5A. Let $p\left(\theta ,\varphi \right)$ represent the probability of a spherical PCN adsorbed on the analyte surface. Consequently, both $G_{SERS}^{0}\left(\theta ,\varphi \right)$ and $h\left(\theta ,\varphi \right)$ become functions of θ and φ. If $M_{PCN}$ denotes the average number of spherical PCNs adsorbed on an analyte particle, $n_{A}$ represents the density of the analyte particles in the solution, and V denotes the volume of the detection (the blue dashed box in Figure 2), then:(31)$$I_{SERS}^{sphere}=F_{AH}M_{PCN}n_{A}I_{0}V\overset{2\pi }{\underset{0}{\int }}d\varphi \overset{\pi }{\underset{0}{\int }}p\left(\theta ,\varphi \right)G_{SERS}^{0}\left(\theta ,\varphi \right)sin\theta d\theta \frac{4}{\delta }\int_{0}^{\infty }\eta_{M}\left(z\right)\sigma_{AH}\left(z\right)e^{-\frac{4z}{\delta }}\pi h{\left(\theta ,\varphi \right)}^{2}dz,$$
where $\int_{0}^{2\pi }d\varphi \int_{0}^{\pi }p\left(\theta ,\varphi \right)sin\theta d\theta =1$. Note that $M_{PCN}$ should represent a function of $n_{A}$ and $n_{PCN}$, where $n_{PCN}$ is the density of spherical PCNs. If the orientation of the surface molecules on the analyte particle surface in the hot spot regions has a distribution, then $G_{sphere}^{0}$ in Equation (31) will be replaced by $G_{sphere}^{D}$, which is determined via Equation (13).

In the case of non-polarized excitation (Figure 5B), the hot spot region forms a band on the PCN, allowing more surface molecules on the analyte particle to contribute to the SERS signal. Due to the symmetry, both $G_{SERS}^{0}\left(\theta \right)$ and $h\left(\theta \right)$ become functions of θ only. Equation (31) remains valid with the modification $I_{0}\to I_{0}/2$; thus:(32)$$I_{SERS}^{sphere}=F_{AH}M_{PCN}I_{0}n_{A}V\int_{0}^{2\pi }d\varphi \int_{0}^{\pi }p\left(\theta ,\varphi \right)G_{SERS}^{0}\left(\theta \right)sin\theta d\theta \frac{2}{\delta }\int_{0}^{\infty }\eta_{M}\left(z\right)\sigma_{AH}\left(z\right)e^{-\frac{4z}{\delta }}\pi h{\left(\theta \right)}^{2}dz.$$

In comparison to the expression for linear polarized excitation, Equation (31), it has been anticipated that non-polarized excitation can significantly enhance the SERS intensity.

If the analyte particle possesses an inhomogeneous surface, as shown in Figure 5, featuring two distinct regions (as seen in bacterial membranes), denoted as I and H, with different surface molecules characterized by their corresponding scattering cross-sections $\sigma_{AH}^{I}$ and $\sigma_{AH}^{H}$, respectively, the situation becomes more complex. Assuming a $P_{I}$ fraction of PCNs adsorbs on region I, and a $P_{H}$ fraction on region H (where $P_{I}+P_{H}=1$), then:(33)$$I_{SERS}=F_{AH}I_{0}M_{PCN}\left(G_{AH}^{I}P_{I}\sigma_{AH}^{I}+G_{AH}^{H}P_{H}\sigma_{AH}^{H}\right)Vn_{A},$$
where $G_{AH}^{I}$ and $G_{AH}^{H}$ denote theoretical EFs of corresponding molecules. Equation (33) shows that, in principle, the overall SERS spectra are a linear combination of $\sigma_{AH}^{I}\left(\Delta v\right)$ and $\sigma_{AH}^{H}\left(\Delta v\right)$ (both could be depth-dependent, as shown in Equation (29)). However, the coefficients in this linear combination do not only rely on $P_{I}$ and $P_{H}$ but also on their corresponding SERS EFs ($G_{SERS}^{DI}$ and $G_{SERS}^{DH}$). If the PCNs are not specifically designed to preferentially bind to any region, $P_{I}/P_{H}=A_{I}/A_{H}$, where $A_{I}$ and $A_{H}$ represent the surface areas of regions I and H on the analyte particle, respectively. If the PCNs are selectively modified via certain chemical functionalization groups, $P_{I}/P_{H}$ will be extremely specific. If the surface of the analyte particle comprises more than two inhomogeneous regions, Equation (33) will present an accumulation of SERS spectra from different surface regions, i.e., Equation (33) can be extended to situations involving three or more surface components.

***Spheroid PCNs*:** If spheroid PCNs are employed as a SERS substrate to detect analyte particles significantly larger than the PCNs (as illustrated in Figure 6A) under linearly polarized light, with $\lambda_{ex}\approx \;\lambda_{LSPR}^{L}$, only PCNs with one of their poles adsorbed on the analyte particle and oriented in alignment with the polarization direction can contribute to the SERS signal. This includes the particles numbered one, six, and seven in Figure 6A.

The probability of the spheroid poles adsorbed on the analyte particle is provided by Equation (32). If there are a total number of $N_{PCN}$ particles on the analyte particle surface, the number of PCN particles that could potentially produce SERS is $M_{PCN}P_{prolate}^{L}$. Only those PCNs adsorbed at locations with a local surface normal component aligned with the polarization direction can generate the SERS signal. Based on Equation (31), one has:(34)$$I_{SERS}^{spheriod-L}=G_{spheriod}^{LD}F_{H}\sigma_{SERS}I_{0}M_{PCN}P_{prolate}^{L}Vn_{A}\Omega^{L},$$
with $\Omega^{L}=\int_{0}^{2\pi }d\varphi \int_{0}^{\pi }p^{L}\left(\theta ,\varphi \right)N_{A}^{ML}\eta^{L}\left(\theta ,\varphi \right)sin\theta d\theta$. For non-polarized excitation, the argument for Equation (32) remains valid, and the total SERS intensity will increase by 4r/h.

A similar argument for inhomogeneous analyte particles is also valid.

***Spherical PCN Dimers*:** If PCN dimer particles are used under linearly polarized excitation (as shown in Figure 6B), it becomes evident that none of the surface molecules of the analyte particle can be located inside the hot spot positions (gaps). Consequently, using $\lambda_{ex}$ to excite the hot spot gap for generating the SERS signal is not advantageous. In the scenario where two PCN spherical particles form a dimer, plasmon hybridization results in two longitudinal modes and one transverse mode [17]. The hot spot gap emerges due to the bonding longitudinal mode with a resonant wavelength, $\lambda_{bonding}^{L}$, while the anti-bonding mode, $\lambda_{anti-bonding}^{L}<\lambda_{bonding}^{L}$, and the transverse mode, $\lambda^{T}<\lambda_{bonding}^{L}$ produce relative weak local electric fields. For $\lambda_{anti-bonding}^{L}$, the hot spots are at the two ends of the dimer along the long-axis direction, whereas for $\lambda^{T},$ the hot spots are at the four tops of the two spheres perpendicular to the dimer’s long axis, as indicated by the dashed rectangle in Figure 6B. Therefore, to generate a sufficient SERS signal, one must choose $\lambda_{ex}\approx \lambda_{anti-bonding}^{L}$ or $\lambda_{ex}\approx \lambda^{T}$. These two cases align precisely with the spheroid PCN situations discussed earlier. It was anticipated that the produced SERS intensities will be determined through Equation (34).

This configuration demonstrates that the hot spot arrangement in a SERS substrate may not necessarily be consistent for different analytes. While the hot spot gap configuration in PCN dimers is useful for explaining SERS signals when analyte molecules are much smaller than the gap size, as the size of analyte molecules becomes comparable or even larger than the gap, the hot spot may shift to different locations on the two spherical PCNs. Consequently, adjustments in $\lambda_{ex}$ are necessary to obtain the maximum SERS intensity.

### 3.2. Film-Based SERS Measurements

Figure 7 presents four distinct types of thin-film SERS substrates, each with unique characteristics that profoundly impact SERS performance. The first type, ultra-thin substrates (Figure 7A), can be prepared using various methods, such as conventional lithography methods, nanosphere lithography, or coating a sub-monolayer of PCNs on the substrate [7,18]. These substrates are typically less than 100 nm thick. The second type, bundling-induced hot spot substrates (Figure 7B), consists of long non-plasmonic nanorods with plasmonic particles coated on their tips. When immersed in a liquid and dried, capillary effects and dewetting cause the nanorods to bundle, creating hot spots at the gaps between the top nanoparticles [19,20,21,22]. The third type, porous SERS substrates (Figure 7C), utilizes a porous inorganic or organic thin film as a host for plasmonic particles dispersed into the pores [23,24,25]. The porous structure can be sol–gel films or fiber networks, and the plasmonic particles can be pre-synthesized, synthesized in situ, or evaporated. The fourth type, porous plasmonic thin films (Figure 7D), consists of pure plasmonic material, such as the silver nanorod substrate fabricated via oblique angle deposition [26]. Multilayer PCN films can also be similarly treated. Due to the significant differences in structure, morphology, and hot spot density among these substrates, they can exhibit extremely diverse SERS performances.

The SERS signal measured can be significantly affected by the method used to prepare analyte samples for thin-film SERS substrates. Two typical methods that are employed are drop-casting and immersion. In drop-casting (illustrated in Figure 8 for ultra-thin and bundle substrates), an analyte solution with a volume of *V_A_* and a concentration of *n*_A_ is dispensed onto the substrate (step 1). The droplet can either spread or remain, depending on the solution’s hydrophilicity/hydrophobicity. Subsequently, as the droplet evaporates and dewets from the substrate surface (step 2), plasmonic particles not firmly attached may be displaced due to capillary forces. An uneven analyte concentration may lead to a coffee-ring effect [27,28]. In the case of the bundle substrate, vertically aligned nanorods initially bundle together during dewetting, forming gap-like hot spots. To prevent non-uniform distribution, confining the droplet within a well on the substrate can ensure even spreading, aiding in a more uniform evaporation. Drop-casting is a non-equilibrium method where adsorption–desorption equilibrium is not reached, depending on the evaporation speed. However, all analytes in the droplet are deposited onto the substrate.

The immersion method involves immersing the substrate in an analyte solution for a specific time to establish an adsorption–desorption equilibrium, followed by drying and subsequent SERS measurement. This method requires time for equilibrium establishment, and, in some cases like bundle substrates, drying is necessary to form hot spots. In subsequent discussions, we focused on dried substrates, excluding the dynamic immersion scenario.

Furthermore, SERS measurements significantly depend on optical configurations, including incident and collection angles and the polarization of the excitation laser. In most configurations, backscattered signals from thin-film substrates are collected under a zero incident angle. Occasionally, the collection configuration remains fixed, while the incident angle varies [29]. The polarization of the excitation laser plays a vital role, influencing SERS signal strength and spectral shape based on substrate morphology and analyte molecule orientation. For anisotropic substrates, like Ag nanorod array (AgNR) substrates, the laser’s polarization strongly impacts the SERS spectrum [30]. Additionally, ultra-thin substrates are susceptible to changes in spectral shape if analyte molecules tend to alter their orientation during adsorption [31].

Finally, as discussed in solution-based detection, the size of analyte molecules or particles significantly impacts SERS measurements, determining their locations within hot spots. Therefore, the discussions below are based on analyte size.

#### 3.2.1. Small Analytes

***Ultra-thin substrates***: As shown in Figure 9, we consider three typical ultra-thin substrates formed by dispersing a sub-monolayer of spherical, spheroid, and dimer-like PCNs on a flat solid substrate (such as glass, Si, or others). Various substrates created through conventional or nonconventional lithography methods can follow the same principles discussed here.

For the substrate formed by spherical PCNs (Figure 9A), as discussed in Section 3.1, with a horizontally polarized excitation, if the theoretical EF for a hot spot is $G_{AH}^{0}$, and the analytes can be randomly adsorbed on each PCN, then Equation (13) holds true by averaging the molecular orientation on the PCNs. In the case of sample preparation through immersion, the results resemble those from solution-based measurements since analyte–PCN absorption reaches equilibrium. Assuming a uniform adsorption of analyte molecules on each PCN with an average count of *M*_A_, either Equation (15) or (16) remains valid. However, for the drop-casting method, the estimation of $G_{ultra-thin}^{A}$ differs. If a volume, $V_{d}$, of analytes with a bulk concentration of $n_{A}$ is dispensed on the substrate, with a spreading area of $A_{s}$, the surface concentration of the analytes becomes:(35)$$\eta_{A}=\frac{n_{A}V_{d}}{A_{s}}.$$

Assuming the surface density of PCNs is $\eta_{PCN}$ and the hot spot density is 2$\eta_{PCN}$, the effective total surface area in the droplet spread area becomes $A_{s}(1$+$4\pi r^{2}\eta_{PCN}),$ and the hot spot area is 2$A_{s}\eta_{PCN}r^{2}\Omega_{H}$. We assumed a uniform probability of analytes adsorbing on both PCN surfaces and substrate surfaces. The average EF is then calculated as follows:(36)$$G_{ultra-thin}^{A}=G_{ultra-thin}^{D}\frac{2\eta_{PCN}r^{2}\Omega_{H}}{1+4\pi r^{2}\eta_{PCN}}.$$

Similarly, if we consider the potential size and aggregation of the PCNs, Equation (18) remains valid.

In immersion measurements, achieving an adsorption–desorption equilibrium between analytes and both PCN surfaces and exposed substrate surfaces is crucial. It is important to note that the adsorption isotherms on these surfaces may not be identical, potentially leading to a different form of Equation (36).

For substrates created with spheroid PCNs (Figure 9B), the approach outlined in Section 3.1.1 and the preceding discussion can be applied. The same holds true for thin-film substrates based on dimer formations (Figure 9C).

***Bundle substrates:*** Assuming that the PCNs on bundle substrates are spherical in shape, each on a cylindrical nanorod with a height of $h_{b}$ and a diameter of $d_{b}$, the average EF can be calculated considering possible orientations upon drop-casting:(37)$$G_{bundle}^{A}=G_{bundle}^{D}\frac{\eta_{PCN}r^{2}\Omega_{H}}{1+\left(4\pi r^{2}+\pi d_{b}h_{b}\right)\eta_{PCN}}.$$

For immersion measurements, where there are three distinct surfaces—PCN, substrate, and nanorod array—the expression for Equation (37) would need to be adjusted accordingly.

***Porous substrates*:** For a porous substrate, let us consider a substrate with a hot spot density of $n_{hs}$, where each hot spot occupies a volume of $V_{hs}$, and the substrate has a thickness of $d_{porous}.$ For the drop-casting method, the actual analyte concentration on the substrate can be written as:(38)$$n\prime_{A-p}=\frac{n_{A}V_{d}}{A_{s}d_{porous}}.$$

The number of analyte molecules on a single hot spot is calculated as:(39)$$M_{A}=V_{hs}n\prime_{A-p}=V_{hs}\frac{n_{A}V_{d}}{A_{s}d_{porous}}.$$

The total number of analyte molecules per hot spot occupied volume is:(40)$$M\prime_{A}=\frac{1}{n_{hs}}n\prime_{A}=\frac{1}{n_{hs}}V_{hs}\frac{n_{A}V_{d}}{A_{s}d_{porous}},$$

Thus, the average EF can be written as:(41)$$G_{porous}^{A}=G_{porous}^{D}n_{hs}V_{hs}.$$

Equation (41) demonstrates that to enhance the effective EF, increasing both hot spot density and hot spot volume is essential.

Up to this point, Equations (35), (36), and (41) provide the formulas for the average EF for different thin-film substrates. Taking into account the variations in diameter and shape of the PCNs within each substrate, an additional averaging process based on shape and size, similar to Equation (18), is necessary across these three equations to derive the effective EF.

#### 3.2.2. Large Analytes

Unlike the scenario in Section 3.1.2, where PCNs can adsorb randomly all over the surface of a large analyte particle, in thin-film substrates, the analyte particle can only rest on the surface of the substrates, as illustrated in Figure 10. Therefore, only the very top portions of the SERS substrates in direct contact with the analyte particle surface can generate SERS signals, constituting the hot spot locations. Consequently, irrespective of the substrate types, the generated SERS signal should exhibit similar behavior.

For each contact point between the SERS substrate and the analyte particle, considering the distance-dependent EF, Equation (29) remains applicable. If each analyte particle has $M_{hs}$ hot spot contact points on the substrate and a surface density of $\eta_{AN}$, then the SERS signal can be expressed as:(42)$$I_{SERS}=\eta_{AN}A_{s}M_{hs}I_{AH}^{S}=F_{AH}I_{0}\eta A_{s}M_{hs}G_{SERS}^{0}\frac{4}{\delta }\int_{0}^{\infty }\eta_{M}\left(z\right)\sigma_{AH}\left(z\right)e^{-\frac{4z}{\delta }}\pi h^{2}dz.$$

Equation (42) shows that regardless of the type of thin-film substrate, the SERS intensity from a large analyte particle is directly proportional to the analyte particle’s surface density, the number of contacts between the SERS substrate and the analyte particle, as well as $G_{SERS}^{0}$. The spectral shape is determined by the integral in Equation (42), representing the depth homogeneity of the analyte particle. If the surface of analyte particles is non-uniform, as explored in Section 3.1.2, the ultimate SERS spectral profile is contingent upon the interaction between the analyte particle and the substrate. This dependence involves factors such as the proportion of various regions on the analyte surface in contact, the associated SERS enhancement factors, and the scattering cross-sections. Consequently, an equation combining Equations (33) and (42) can be derived to encapsulate these influences.

In thin-film substrate cases, even if a SERS signal is obtained, it would be significantly smaller compared to solution-based detection (and under a similar PCN configuration, as shown in Figure 5 and Figure 6). This decrease in the SERS signal arises from two primary reasons: firstly, since the collection configuration involves backscattering with a zero incident angle, hot spots only occur in the horizontal direction; and secondly, even if hot spots occasionally form on top of the substrates, $G_{SERS}^{0}$ would be considerably smaller than that in actual hot spots. It is intriguing to explore how the ideal EF $G_{SERS}^{0}$ can be generated under the conventional backscattering measurement configuration shown in Figure 7, given that the polarization of the incident excitation laser beam is always parallel to the thin film’s surface.

For an ultra-thin film composed of spherical PCNs, as shown in Figure 11A, when excited by a normally incident laser beam, the hot spots emerge on the horizontal side surfaces of each spherical PCN, which cannot come into contact with an analyte particle. Consequently, to ensure the hot spot contacts the analyte particles, the incident laser configuration must be altered, specifically by introducing a particular angle of *θ*, as shown in Figure 11B. A large *θ* (i.e., close to 90°) allows for a larger hot spot volume to interact with the analyte particle, thereby generating a larger $G_{SERS}^{0}$. However, changing a commercial system’s optical configuration from normal incidence to grazing incidence is challenging.

Another thin-film configuration involves using aligned spheroid PCN particles, as shown in Figure 11C. When these spheroid PCNs are vertically aligned, under the normal incident configuration, hot spots only form on the horizontal side surfaces of the PCNs near the transverse mode, $\lambda_{LSPR}^{T}$ (see Section 3.1.1). However, if the aligned spheroid PCNs are tilted at an angle of β with respect to the surface normal, as shown in Figure 11D, both the longitudinal mode, $\lambda_{LSPR}^{L}$, and the transverse mode, $\lambda_{LSPR}^{T}$, can be excited for the PCN array. Notably, when the longitudinal mode is excited, the hot spot will form at the tip of each PCN. Thus, it is expected that a higher SERS signal will be produced. Eventually, the larger the β, the greater the $G_{SERS}^{0}$, and the higher the hot spot volume. This discussion illustrates the significant impact of the SERS measurement’s optical configuration on the measured SERS intensity.

## 4. Optical Attenuation during the SERS Signal Collection

The discussion above has focused on how the SERS signal may be influenced by the effective EF resulting from potential interactions between the analyte and the SERS substrate, as well as the excitation polarization. However, during SERS measurements, both the excitation laser and the SERS signal must travel through the analyte–SERS substrate system. This implies that the effective optical properties of the analyte–SERS substrate system could significantly impact the final collected SERS signals, contributing to $R_{SERS}$ and $R_{R}$ in Equations (10) and (11), respectively. As the optical responses differ between solution-based measurements and thin film-based measurements, we will discuss the effects of excitation laser propagation and attenuation based on these two measurement configurations.

### 4.1. Solution-Based Measurements

As shown in Figure 1, both the excitation light and the collected SERS signal must travel a specific distance in the solution in order to excite the valid PCN volume and to be collected by the instrument. Thus, both the intensity of the excitation laser and the collected scattered light can be attenuated via the optical absorption of the solution or suspension. In Figure 1, at location *z*, the excitation laser intensity will be attenuated to be:(43)$$I\left(z\right)=e^{-\alpha^{ex}z}I_{0},$$
where the superscript “*ex*” indicates a quantity at $\lambda_{ex}$, meaning $\alpha^{ex}=\alpha \left(\lambda_{ex}\right)$ represents the optical absorption coefficient of the measured liquid system at the wavelength of $\lambda_{ex}$. The emitted SERS intensity from location *z* is also attenuated by $e^{-\alpha z}$, where $\alpha =\alpha \left(\Delta v\right)$ is the optical absorption coefficient of the measured liquid at any given Raman shift Δv relative to $\lambda_{ex}$. Hence, according to Equation (8), the SERS signal collected from a *dz* layer can be written as (assuming that the analytes are much smaller than the size of the PCNs):(44)$$dI\prime_{SERS}\left(\Delta v\right)=G_{SERS}^{e}F_{H}\sigma_{AH}A\left(z\right)n_{PCN}M_{A}e^{-\alpha^{ex}z}e^{-\alpha z}I_{0}dz,$$
where $A\left(z\right)$ is the area of the laser beam at location *z*, and the total SERS intensity received by the SERS instrument is:(45)$$I\prime_{SERS}\left(\Delta v\right)=\int_{f-d}^{f+d}G_{SERS}^{e}F_{H}\sigma_{AH}A\left(z\right)n_{PCN}M_{A}e^{-\alpha^{ex}z}e^{-\alpha z}I_{0}dz.$$

Considering the focused excitation laser is a Gaussian beam with a minimum waist, $w_{0}$, at the focal point, the waist $w\left(z\right)$ can be written as:(46)$$w\left(z\right)=w_{0}\sqrt{1+{\left[\frac{\lambda_{ex}\left(z-f\right)}{\pi w_{0}^{2}}\right]}^{2}}.$$

Thus, $A\left(z\right)$ can be approximated by $A\left(z\right)=\pi w{\left(z\right)}^{2}=\pi w_{0}^{2}+\lambda_{ex}^{2}{\left(z-f\right)}^{2}$. Since in most cases $\lambda_{ex}\ll w_{0}$ and if d≪f, $A\left(z\right)\approx \pi w_{0}^{2}$, we have:(47)$$I\prime_{SERS}\left(\Delta v\right)=\frac{G_{SERS}^{e}F_{H}\sigma_{AH}n_{PCN}M_{A}I_{0}\pi w_{0}^{2}}{\alpha_{l}}e^{-\alpha_{l}f}\left[e^{\alpha_{l}d}-e^{-\alpha_{l}d}\right]\approx 2dG_{SERS}^{e}F_{H}\sigma_{AH}n_{PCN}M_{A}I_{0}\pi w_{0}^{2}e^{-\alpha_{l}f},$$
where $N_{A}=2\pi w_{0}^{2}dn_{PCN}M_{A}$, $\alpha_{l}\left(\Delta v\right)=\alpha^{ex}+\alpha \left(\Delta v\right)$, and $\alpha_{l}d\ll 1$. Thus, $R_{SERS}=e^{-\alpha_{l}f}$. Equation (47) indicates that the overall SERS intensity experiences attenuation by $e^{-\alpha_{l}f}$, i.e., by both $\alpha_{ex}$ and $\alpha \left(\Delta v\right)$. If $\alpha \left(\Delta v\right)=0$, $\alpha^{ex}$ results in a constant attenuation across the entire SERS spectrum, preserving the SERS spectrum’s features while reducing their intensity by a factor of $e^{-\alpha^{ex}f}$. This reduction has been considered in estimating the actual measured SERS EF $G_{m}$ according to Equation (12). If $\alpha^{ex}=0$, $\alpha \left(\Delta v\right)$ alters the shape of the SERS spectrum, causing distortion from the true SERS spectrum since the attenuation at different SERS shifts (Δv) is different.

According to Figure 1, both $\alpha^{ex}$ and $\alpha \left(\Delta v\right)$ can arise from three potential sources: First, the optical absorption of un-adsorbed analytes in the solution with a concentration of $n\prime_{A}$, $n\prime_{A}=n_{A}-n_{PCN}M_{A},$ following the Beer–Lambert law:(48)$$\alpha_{A}^{ex}=\varepsilon_{A}^{ex}n\prime_{A}\;\mathrm{and}\;\alpha_{A}\left(\Delta v\right)=\varepsilon_{A}\left(\Delta v\right)\;n\prime_{A},$$
where $\varepsilon_{A}$ is the absorptivity of a single analyte in the solution, and $\varepsilon_{A}^{ex}=\varepsilon \left(\lambda_{ex}\right)$. In Figure 12A, if $\varepsilon_{A}\left(\Delta v\right)$ exhibits a featureless profile, $\alpha_{A}\left(\Delta v\right)$ will also lack features, leading to nonlinear attenuation across different Δv. Moreover, if $\alpha_{A}$ demonstrates a strong dependence on $n\prime_{A}$ (or $n_{A}$), the SERS intensity, $I_{SERS}$, will systematically change with $n\prime_{A}$ (or $n_{A}$). However, if $\varepsilon_{A}\left(\Delta v\right)$ displays sharp peaks due to intrinsic resonance absorption of the analyte molecules within the SERS wavelength range, these peaks or dips in $\varepsilon_{A}\left(\Delta v\right)$ will significantly attenuate the original SERS spectrum, introducing false features in the measured SERS spectrum.

Second, since in most measurements, to maximize the SERS signal, one typically chooses $\lambda_{ex}\;\approx \lambda_{LSPR}$ to excite the PCNs, as shown in Figure 12A. The PCNs present a strong Δv dependence absorption spectrum, $\alpha_{LSPR}\left(\Delta v\right)$, in the vicinity of $\lambda_{ex}$. The spectral shape is influenced by the size, shape, aggregation of the PCNs used, and the PCN concentration, $n_{PCN}$, as follows:(49)$$\alpha_{LSPR}\left(\Delta v\right)=\varepsilon_{PCN}\left(\Delta v\right)n_{PCN},$$
where $\varepsilon_{PCN}\left(\Delta v\right)$ represents the absorptivity of a single PCN particle in the solution. During the SERS measurement, $n_{PCN}$ remains constant, while the SERS measurement wavelength region aligns with the LSPR resonance region. Consequently, $\alpha_{LSPR}\left(\Delta v\right)$ significantly attenuates the SERS spectrum. However, as analytes adsorb on the PCNs, $\alpha_{LSPR}\left(\Delta v\right)$ will be slightly modified, which can be treated with an effective medium theory and will be discussed in Section 5.

Third, if the analyte is not in an aqueous solution but is in a specific buffer, the optical absorption of the buffer solution also contributes to both $\alpha^{ex}$ and $\alpha \left(\Delta v\right)$, with:(50)$$\alpha_{bf}^{ex}=\varepsilon_{bf}^{ex}n_{bf}\;\mathrm{and}\;\alpha_{bf}\left(\Delta v\right)=\varepsilon_{bf}\left(\Delta v\right)n_{bf}.$$

Here, $\alpha_{bf}$ depends on the concentration of the buffer ($n_{bf}$). If an analyte solution in a buffer is diluted by a solvent, both $n_{A}$ and $n_{bf}$ change simultaneously and could significantly distort the SERS spectrum. Considering all these contributions to $\alpha_{l}$, the final spectral shape of $\alpha_{l}$ could resemble the red curve in Figure 12B. If the SERS excitation wavelength, $\lambda_{ex}$, is selected in different spectra regions, the shape of $\alpha_{l}\left(\Delta v\right)$ to attenuate the SERS spectrum will vary. For example, if the $\lambda_{ex}$ of the 1st, 2nd, 3rd, and 4th locations labeled in Figure 12B are selected, the corresponding $\alpha_{l}\left(\Delta v\right)$ will represent four typical situations, as illustrated in Figure 13A: Case 1, a monotonically decreased $\alpha_{l}$ with respect to Δv; Case 2, a monotonically increased $\alpha_{l}$ with respect to Δv; Case 3, a dip-shaped $\alpha_{l}$ (centered at Δv=1000 cm^−1^); and Case 4, a peak-shaped $\alpha_{l}$ (centered at Δv=1000 cm^−1^). Figure 13B shows an experimentally obtained SERS spectrum, $I_{SERS}\left(\Delta v\right),$ of trans-1,2-Bis(4-pyridyl)ethene (BPE), treated as a standard and original SERS spectrum. This SERS spectrum will be multiplied by $e^{-\alpha_{l}}$ for cases 1–4 to demonstrate the spectral distortion, $I\prime_{SERS}\left(\Delta v\right)$.

Figure 13C–F show the resulting $I\prime_{SERS}\left(\Delta v\right)$. In Case 1 (Figure 13C), more absorptions occurred in the low Δv region, leading to the suppression of relative spectral intensities of $I_{SERS}\left(\Delta v\right)$ at low Δv and enhancement at high Δv. Conversely, in Case 2 (Figure 13D), the opposite trend was observed: the relative intensities at the low Δv region are enhanced, and the overall spectral intensity significantly decreases due to the high absorbance. In these two cases, the attenuations are small, making it visually challenging to discern obvious spectral shape differences between $I_{SERS}\left(\Delta v\right)$ and $I\prime_{SERS}\left(\Delta v\right)$. For Case 3 (Figure 13E), the spectral shape of $I\prime_{SERS}\left(\Delta v\right)$ appears to be significantly different from $I_{SERS}\left(\Delta v\right)$: the peak intensity at Δv=1206 cm^−1^ becomes the maximum peak in $I\prime_{SERS}\left(\Delta v\right)$, while in $I_{SERS}\left(\Delta v\right)$ (Figure 13B) the maximum intensity peak is at Δv=1616 cm^−1^. This discrepancy arises as absorption attenuation enhances the peak intensities near Δv=1000 cm^−1^, due to the dip in $\alpha_{l}\left(\Delta v\right)$. Case 4 (Figure 13F) shows opposite results; the peak intensities near Δv=1000 cm^−1^ were suppressed, while the peak intensities at the two edges were enhanced. Notably, the peak intensities at Δv=1616 cm^−1^ and Δv=1646 cm^−1^ were nearly identical, unlike other spectra where the intensity at Δv=1616 cm^−1^ is consistently larger than that at Δv=1646 cm^−1^. Clearly, the optical properties of the solution can significantly distort the measured SERS spectrum and alter the relative ratios of the peak intensities. It is evident that such distortions can be modified by selecting different $\lambda_{ex}$ to measure the same targeted analyte system.

In addition, if $\alpha_{l}$ is closely linked to $n_{A}$ (or $n_{bf}$), changes in $n_{A}$ can distort the SERS spectrum differently. Let us consider Case 3 and assume that $\alpha_{l}\propto \;n_{A}$. Figure 14A plots $\alpha_{l}$, $2\alpha_{l}$, $2.5\alpha_{l}$, and $3\alpha_{l}$, representing varying $n_{A}$. All four curves in Figure 14A exhibit a dip centered at Δv=1000 cm^−1^. The increased coefficient in front of $\alpha_{l}$ shows an increase in the concentration of $n_{A}$. After multiplying $I_{SERS}\left(\Delta v\right)$ by $e^{-\alpha_{l}}$, $e^{-2\alpha_{l}}$, $e^{-2.5\alpha_{l}}$, and $e^{-3\alpha_{l}}$, respectively, the resulting normalized $I\prime_{SERS}\left(\Delta v\right)$ is plotted in Figure 14B. These spectra do not overlap; instead, with increasing $n_{A}$, the normalized peaks at Δv=1017 cm^−1^ and 1206 cm^−1^ increased, while the peaks at Δv=1616 cm^−1^ and 1646 cm^−1^ decreased. This systematic distortion demonstrates that the distorted spectrum’s shape contains $n_{A}$ information. This forms the theoretical basis for using normalized SERS spectra in machine learning and deep learning regression and classification models to predict the concentration of $n_{A}$.

Figure 13 and Figure 14 also show that when calculating $G_{SERS}^{e}$, based on Equation (12), even for the same Raman molecule, using different SERS peaks may result in different $G_{SERS}^{e}$ due to absorption-induced spectral distortion. They also demonstrate that at different Raman molecule concentrations, for the same SERS peak, the obtained $G_{SERS}^{e}$ can present a function of $n_{A}$.

Clearly, to experimentally obtain the true SERS spectrum, $I_{SERS}\left(\Delta v\right)$, both $I\prime_{SERS}\left(\Delta v\right)$ and $\alpha_{l}\left(\Delta v\right)$ of the target analyte–substrate system should be measured. Based on Equation (47), $I_{SERS}\left(\Delta v\right)=I\prime_{SERS}\left(\Delta v\right)\;e^{\alpha_{l}f}$. This correction can yield a standard SERS spectrum of the target analyte.

If the analyte particle is much larger than the size of the PCNs, like the situation in Section 3.2, based on the argument for Equation (47), a similar optical response function $R_{SERS}$ will be found for Equation (31), i.e.,:(51)$$I\prime_{SERS}^{sphere}\left(\Delta v\right)=I_{SERS}^{sphere}R_{SERS}=I_{SERS}^{sphere}e^{-\alpha_{l}f}.$$

Clearly, the measured SERS spectrum $I\prime_{SERS}^{sphere}\left(\Delta v\right)$ is also distorted by $\alpha_{l}\left(\Delta v\right)$. Note that $\alpha_{l}\left(\Delta v\right)=\alpha^{ex}+\alpha \left(\Delta v\right).$ In this situation, all the above discussions hold true. However, the factors contributing to $\alpha_{l}\left(\Delta v\right)$ become more intricate. There are four potential sources contributing to $\alpha_{l}$: the freely suspended PCNs in the solution, which contribute to LSPR-like extinction, $\alpha_{LSPR}$; the freely suspended analyte particles, leading to extinction due to particle scattering, $\alpha_{A}$; the hybrid PCN–analyte particle system, as shown in Figure 5, Figure 6 and Figure 7, which may introduce a complicated optical response, $\alpha_{hybrid}$; and finally, the possible contribution from the buffer solution, $\alpha_{bf}$. Unlike the small-size analyte situation, estimating both $\alpha_{A}$ and $\alpha_{hybrid}$ could be very complicated. An analyte particle can be treated as a homogenous or inhomogeneous dielectric particle, requiring the exploration of the Mie scattering theory to estimate $\alpha_{A}$ since its size is comparable or even larger than $\lambda_{ex}$, and its shape can vary [32]. The case for a PCN–analyte particle is even more complicated, since it is an inhomogeneous particle with a distribution of the number of PCNs on an analyte particle. $\alpha_{hybrid}$ can be estimated based on an approximation using an effective particle through the Mie theory [32] or via numerical calculations.

### 4.2. Thin-Film-Based Measurements

For thin-film-based SERS substrates, there are typically two interfaces, and occasionally three or four, between the air and the substrate, or between the plasmonic layer and another dielectric layer. When examining the overall SERS intensity, one must account for these interfaces. During the propagation of the excitation laser and collection of the SERS signal, the impact of multiple interfacial reflections and transmissions, as well as propagation attenuation effects, must be taken into account. These complexities make the final collected SERS signal extremely complicated. In the following discussion, we will focus on situations involving drop-casting on three specific substrates: ultra-thin films, bundled thin films, and porous thin films.

#### 4.2.1. The Ultra-Thin Substrates

In the case of the ultra-thin-film substrate, the monolayer SERS substrate can be considered as an effective layer, denoted as 2 in Figure 15A. The excitation laser reflects at the 1–2 and 2–3 interfaces, resulting in the actual excited laser intensity, which is the sum of the first transmitted intensity at the 1–2 interface and the reflected intensity at the 2–3 interface:(52)$$I_{ex}=I_{0}T_{12}^{ex}\left[1+e^{-\alpha_{2}^{ex}d_{2}}T_{23}^{ex}\right]=I_{0}T^{ex},$$
where $T^{ex}=T_{12}^{ex}\left[1+e^{-\alpha_{2}^{ex}d_{2}}T_{23}^{ex}\right]$ represents a SERS intensity modulation factor when $\lambda_{ex}$ is fixed. The collected SERS signal comprises two components: the signal directly from the hot spot transmitted via the 2–1 interface, and the SERS signal reflected from the 2–3 interface and then transmitted through the 2–1 interface:(53)$$I\prime_{SERS}\propto T_{21}\left[e^{-\frac{\alpha_{2}\left(\Delta v\right)d_{2}}{2}}+e^{-\frac{3\alpha_{2}\left(\Delta v\right)d_{2}}{2}}R_{23}\right].$$

Therefore, the total SERS signal can be expressed as:(54)$$I\prime_{SERS}=I_{AH}R_{SERS}=I_{AH}T^{ex}T_{21}\left[e^{-\frac{\alpha_{2}\left(\Delta v\right)d_{2}}{2}}+e^{-\frac{3\alpha_{2}\left(\Delta v\right)d_{2}}{2}}R_{23}\right],$$
where $\alpha_{2}$ is the effective absorption coefficient of the ultra-thin-film substrate, and $I_{AH}$ is defined through Equation (2), representing the SERS intensity without considering the optical response of the collection. Given that the SERS signal is collected using a backscattering configuration with a zero incident angle, the transmission $T_{if}$ and reflectance $R_{if}$ (where *i* indicates the incident medium, and *f* represents the refractive medium) follow the Fresnel equations:(55)$$\left\{\begin{array}{c} T_{if}={\left|\frac{2\xi_{i}}{\xi_{i}+\xi_{f}}\right|}^{2}\\ R_{if}={\left|\frac{\xi_{i}-\xi_{f}}{\xi_{i}+\xi_{f}}\right|}^{2} \end{array}\right.,$$
where $\xi_{i}$ and $\xi_{f}$ denote the complex (effective) indices of refraction of the *i* and *f* layers, respectively. Assuming that the adsorption of analytes does not significantly alter the optical property of the PCN layer in Figure 15, $T_{12}$, $T_{21}$, and $R_{23}$ can be treated as constants.

Now, let us estimate $\alpha_{2}$, which is a combined effect of the PCN layer and the adsorbed analytes. The ultra-thin layer can be treated as an effective layer with the PCNs and analytes. Let the dielectric functions of these two materials be:(56)$$\left\{\begin{array}{c} \varepsilon_{p}=\varepsilon_{pr}+i\varepsilon_{pi}\\ \varepsilon_{A}=\varepsilon_{Ar}+i\varepsilon_{Ai} \end{array}\right.,$$
where $\varepsilon_{pr}$ and $\varepsilon_{Ar}$ are the real parts and $\varepsilon_{pi}$ and $\varepsilon_{Ai}$ are the imaginary parts of materials for the PCNs and analytes, respectively. Assuming a uniform spread of analyte solution with a volume of $V_{s}$ and a concentration of $n_{A}$ on a surface area, $A_{s}$, of a SERS substrate, the volume fraction, $\delta_{2A}$, of analytes on the substrate can be calculated as:(57)$$\delta_{2A}=\frac{V_{A}V_{s}n_{A}}{A_{s}d_{2}}.$$

When the analyte concentration is low, causing minimal perturbation in the optical response of the system, the effective dielectric function, $\varepsilon_{eff}$, can be estimated according to the Maxwell–Garnett theory [33]:(58)$$\varepsilon_{eff}=\varepsilon_{p}\frac{2\delta_{2A}\left(\varepsilon_{A}-\varepsilon_{p}\right)+\varepsilon_{A}+2\varepsilon_{p}}{\varepsilon_{A}+2\varepsilon_{p}-\delta_{2A}\left(\varepsilon_{A}-\varepsilon_{p}\right)}.$$

If $\delta_{2A}\ll 1$ and $\left|\varepsilon_{p}\right|\gg \left|\varepsilon_{A}\right|$ (since the PCN layer is usually made of noble metals), Equation (58) can be rewritten as:(59)$$\varepsilon_{eff}\approx \left(1-\delta_{2A}\right)\varepsilon_{p}+\delta_{2A}\varepsilon_{A}=\varepsilon_{r}+i\varepsilon_{i},$$
with $\varepsilon_{r}=\left(1-\delta_{2A}\right)\varepsilon_{pr}+\delta_{2A}\varepsilon_{Ar}$ and $\varepsilon_{i}=\left(1-\delta_{2A}\right)\varepsilon_{pi}+\delta_{2A}\varepsilon_{Ai}$. $\varepsilon_{eff}={\left(\xi_{eff}+i\kappa_{eff}\right)}^{2}=\xi_{eff}^{2}-\kappa_{eff}^{2}+i2\xi_{eff}\kappa_{eff}$, where $\xi_{eff}$ and $\kappa_{eff}$ are the real and imaginary parts of the effective index of refraction, respectively. Thus:(60)$$\kappa_{eff}=\frac{1}{\sqrt{2}}{\left[-\varepsilon_{r}+{\left(\varepsilon_{r}^{2}+\varepsilon_{i}^{2}\right)}^{\frac{1}{2}}\right]}^{\frac{1}{2}}\approx \frac{1}{\sqrt{2}}{\left(\varepsilon_{1}+\delta_{2A}\varepsilon_{2}\right)}^{1/2},$$
where $\varepsilon_{1}=\left(\varepsilon_{p}-\varepsilon_{pr}\right)$, $\varepsilon_{2}=\varepsilon_{Ar}+\varepsilon_{pr}-\varepsilon_{p}+\frac{\varepsilon_{Ar}\varepsilon_{pr}+\varepsilon_{Ai}\varepsilon_{pi}}{\varepsilon_{p}}$, and $\varepsilon_{p}=\sqrt{\varepsilon_{pr}^{2}+\varepsilon_{pi}^{2}}$. According to Beer–Lambert law:(61)$$\alpha_{2}=\alpha_{eff}=\frac{4\pi \kappa_{eff}}{\lambda }\approx \frac{4\pi }{\lambda_{ex}\sqrt{2}}\left(\sqrt{\varepsilon_{1}}+\frac{\varepsilon_{2}}{2\sqrt{\varepsilon_{1}}}\delta_{2A}\right)=\alpha_{2}^{p}+\alpha_{2}^{A}n_{A}.$$

Here, we let $\lambda =\lambda_{ex}$ since the SERS wavenumber shift is small compared to the excitation wavelength. $\alpha_{2}^{p}=\frac{4\pi \sqrt{\varepsilon_{p}-\varepsilon_{pr}}}{\lambda_{ex}\sqrt{2}}$ is solely dependent on the optical property of the SERS substrate, while $\alpha_{2}^{A}=\frac{\sqrt{2}\pi \varepsilon_{2}}{\lambda_{ex}\sqrt{\varepsilon_{1}}}\frac{V_{A}V_{s}}{A_{s}d_{2}}$ is determined by multiple factors, such as the optical properties of the SERS substrates and the analytes, and the spreading of the analyte on the SERS substrate. Equation (54) changes to:(62)$$I\prime_{SERS}=I_{AH}T^{ex}T_{21}\left[e^{-\frac{1}{2}\left(\alpha_{2}^{p}+\alpha_{2}^{A}n_{A}\right)d_{2}}+e^{-\frac{3}{2}\left(\alpha_{2}^{p}+\alpha_{2}^{A}n_{A}\right)d_{2}}R_{23}\right].$$

From Equation (62), three important conclusions can be drawn: First, in addition to the modification we discussed in Section 3 regarding the SERS EF, the propagation of the excitation laser within the SERS substrates and across different interfaces can further impact the determination of the effective EF. Second, the shape of the SERS spectrum will be significantly influenced by the optical property of the SERS substrate, particularly due to terms such as $e^{-\frac{1}{2}\alpha_{2}^{p}d_{2}}$ and $e^{-\frac{3}{2}\alpha_{2}^{p}d_{2}}$ in Equation (62). Finally, the quantitative relationship between the SERS intensity, $I_{SERS}$, and the analyte concentration,$\;n_{A}$, is extremely complicated. Not only does $I_{AH}$ depend on how the analytes are adsorbed onto the hot spot locations, but it also experiences additional modifications due to terms involving $e^{-\frac{1}{2}\alpha_{2}^{A}n_{A}d_{2}}$ and $e^{-\frac{3}{2}\alpha_{2}^{A}n_{A}d_{2}}$. This indicates that not all SERS peaks will follow the same $I_{SERS}-n_{A}$ relationship. Moreover, for SERS imaging or multi-location measurements using such thin-film-based SERS substrates, if the substrate is nonhomogeneous, leading to varying hot spot densities and local optical properties in different locations, significant variations can occur in the measured SERS spectra.

#### 4.2.2. The Bundle Substrates

For the bundle-like substrate, it can be treated as two effective layers, denoted as layer 2 and layer 3, as shown in Figure 15B. The actual intensity of the excited laser is a combination of the first transmitted intensity at the 1–2 interface, the reflected intensity at the 2–3 interface, and the reflected intensity at the 3–4 interface:(63)$$I_{ex}=I_{0}T^{ex}=I_{0}T_{12}^{ex}\left(1+e^{-\alpha_{2}^{ex}d_{2}}R_{23}^{ex}+e^{-\alpha_{2}^{ex}d_{2}}T_{23}^{ex}R_{34}^{ex}T_{32}^{ex}e^{-2\alpha_{3}^{ex}d_{3}}\right).$$

The SERS signal originates from three sources: the signal directly emerging from the hot spots and transmitted via the 2–1 interface, the SERS signal reflected from the 2–3 interface and passing through the 2–1 interface, and the SERS signal transmitted through the 2–3 interface, propagated through layer 3, and reflected at the 3–4 interface:(64)$$I\prime_{SERS}\propto T_{21}\left(e^{-\frac{\alpha_{2}d_{2}}{2}}+e^{-\frac{3\alpha_{2}d_{2}}{2}}R_{23}+e^{-\frac{3\alpha_{2}d_{2}}{2}}T_{23}R_{34}T_{32}e^{-2\alpha_{3}d_{3}}\right).$$

Therefore:(65)$$R_{SERS}=T^{ex}T_{21}e^{-\frac{\alpha_{2}d_{2}}{2}}\left(1+e^{-\alpha_{2}d_{2}}R_{23}+e^{-\alpha_{2}d_{2}}T_{23}R_{34}T_{32}e^{-2\alpha_{3}d_{3}}\right).$$

Following the earlier discussion, the transmission and reflectance parameters $T^{ex}$, $T_{21}$, $T_{23}$, $T_{32}$, $R_{23}$, and $R_{34}$ can all be considered constants. The estimations of $\alpha_{2}$ and $\alpha_{3}$ can use the effective medium theory based on Equations (57)–(61). However, the estimation of $\delta_{A}$ will differ as there are two porous layers: one is the PCN layer, and the other is the nanorod layer. The quantity of analytes adsorbed on these two layers is proportional to their respective surface areas, assuming uniform adsorption. Let the volume fractions of analytes in layers 2 and 3 be denoted as:(66)$$\delta_{2A}=\beta_{2}n_{A}\;\mathrm{and}\;\delta_{3A}=\beta_{3}n_{A},$$
and the dielectric function for the nanorod layer can be written as $\varepsilon_{d}=\varepsilon_{dr}+i\varepsilon_{di}$. Then, based on the derivations in Equations (57)–(61), $\alpha_{2}$ follows Equation (61), with $\alpha_{2}^{A}=\frac{\sqrt{2}\pi \varepsilon_{2}}{\lambda_{ex}\sqrt{\varepsilon_{1}}}\beta_{2}$, and $\alpha_{3}$ can be written as:(67)$$\alpha_{3}=\alpha_{3}^{d}+\alpha_{3}^{A}n_{A},$$
with $\alpha_{3}^{d}=\frac{4\pi \sqrt{\varepsilon_{d}-\varepsilon_{dr}}}{\lambda_{ex}\sqrt{2}}$, $\alpha_{3}^{A}=\frac{\sqrt{2}\pi \left(\varepsilon_{Ar}\varepsilon_{d}+\varepsilon_{dr}\varepsilon_{d}-\varepsilon_{d}+\varepsilon_{Ar}\varepsilon_{dr}+\varepsilon_{Ai}\varepsilon_{di}\right)}{\lambda_{ex}\varepsilon_{d}\sqrt{\varepsilon_{d}-\varepsilon_{dr}}}\beta_{3}$, and $\varepsilon_{d}=\sqrt{\varepsilon_{dr}^{2}+\varepsilon_{di}^{2}}$. Therefore, Equation (65) becomes:(68)$$I\prime_{SERS}=I_{AH}T^{ex}T_{21}e^{-\frac{\left(\alpha_{2}^{p}+\alpha_{2}^{A}n_{A}\right)d_{2}}{2}}\left[1+e^{-\left(\alpha_{2}^{p}+\alpha_{2}^{A}n_{A}\right)d_{2}}R_{23}+e^{-\left(\alpha_{2}^{p}+\alpha_{2}^{A}n_{A}\right)d_{2}}T_{23}R_{34}T_{32}e^{-2\left(\alpha_{3}^{d}+\alpha_{3}^{A}n_{A}\right)d_{3}}\right].$$

Equation (68) reveals that the total SERS intensity is not only influenced by the optical characteristics of the plasmonic layer but also by the nanorod layer. Consequently, experimentally determining the EF becomes even more complicated. Additionally, the SERS spectrum is altered by the optical properties of both the plasmonic and nanorod layers. This further complicates the $I_{SERS}$–$n_{A}$ relationship, as it relies on the optical properties of both layers.

#### 4.2.3. The Porous Substrates

The porous substrate can be treated as a single effective layer, denoted as layer 2 in Figure 15C. The actual intensity of the laser at position z is a combination of two components: the initial transmitted intensity at the 1–2 interface, and the reflected intensity at the 2–4 interface:(69)$$I_{ex}\left(z\right)=I_{0}T_{12}^{ex}\left(e^{-\alpha_{2}^{ex}z}+e^{-\alpha_{2}^{ex}d_{2}}R_{24}^{ex}e^{-\alpha_{2}^{ex}\left(d_{2}-z\right)}\right),$$
and the SERS signal collected at position *z* with a thickness of *dz* originates directly from the hot spots and is transmitted via the 2–1 interface, as well as from the SERS signal reflected from the 2–4 interface and passing through the 2–1 interface:(70)$$dI\prime_{SERS}=G_{SERS}^{e}F_{H}\sigma_{AH}A_{l}n_{H}M_{A}I_{ex}\left(z\right)T_{21}\left[e^{-\alpha_{2}z}+e^{-\alpha_{2}d_{2}}e^{-\alpha_{2}\left(d_{2}-z\right)}R_{24}\right]dz,$$
where $A_{l}$ is the laser beam area, and $n_{H}$ is the hot spot density. Thus:(71)$$I\prime_{SERS}=G_{SERS}^{e}F_{H}\sigma_{AH}A_{l}n_{H}M_{A}T_{21}T_{12}^{ex}I_{0}\overset{d_{2}}{\underset{0}{\int }}\left(e^{-\alpha_{2}^{ex}z}+e^{-\alpha_{2}^{ex}d_{2}}R_{24}^{ex}e^{-\alpha_{2}^{ex}\left(d_{2}-z\right)}\right)\left[e^{-\alpha_{2}z}+e^{-\alpha_{2}d_{2}}e^{-\alpha_{2}\left(d_{2}-z\right)}R_{24}\right]dz.$$

The integration in Equation (71) provides a rather complicated expression:(72)$$I\prime_{SERS}=G_{SERS}^{e}F_{H}\sigma_{AH}A_{l}n_{H}M_{A}T_{21}T_{12}^{ex}I_{0}\left[\frac{1-e^{-(\alpha_{2}^{ex}+\alpha_{2})d_{2}}}{\alpha_{2}^{ex}+\alpha_{2}}+R_{24}e^{-2\alpha_{2}d_{2}}\frac{1-e^{-(\alpha_{2}^{ex}-\alpha_{2})d_{2}}}{\alpha_{2}^{ex}-\alpha_{2}}-R_{24}^{ex}e^{-2\alpha_{2}^{ex}d_{2}}\frac{1-e^{-(\alpha_{2}-\alpha_{2}^{ex})d_{2}}}{\alpha_{2}^{ex}-\alpha_{2}}+\;R_{24}^{ex}R_{24}e^{-2(\alpha_{2}^{ex}+\alpha_{2})d_{2}}\frac{e^{(\alpha_{2}^{ex}+\alpha_{2})d_{2}}-1}{\alpha_{2}^{ex}+\alpha_{2}}\right].$$

Equation (72) indicates that the SERS intensity of a porous substrate is significantly affected by the substrate’s optical properties. Once the optical characteristics of the porous substrate are determined, the calculation of $\alpha_{2}$ can be conducted using the derivations from Equation (56) to Equation (61).

#### 4.2.4. Large Analyte Particles

When the analyte particles are significantly larger, as shown in Figure 10, the entire sample can be regarded as a four-layer thin-film system, as illustrated in Figure 15D. The analyte particle layer can be treated as a dielectric layer (layer 2) with a thickness of $d_{A}$ and an absorption coefficient of $\alpha_{A}$, while the SERS active layer is considered as layer 3 with a thickness of $d_{p}$ and an absorption coefficient of $\alpha_{p}$. Referring to the discussion in Section 4.2.2, the real excitation intensity consists of three parts, as illustrated in Figure 15D:(73)$$I_{ex}=I_{0}T_{12}^{ex}e^{-\alpha_{A}^{ex}d_{A}}\left(1+R_{23}^{ex}+T_{23}^{ex}R_{34}^{ex}T_{32}^{ex}e^{-2\alpha_{p}^{ex}d_{p}}\right).$$

The SERS signal has two contributions: the direct SERS signal from the interface propagating through layer 2, and the reflected SERS signal at the 3–4 interface. Therefore, the SERS signal can be expressed as:(74)$$I\prime_{SERS}\propto T_{21}e^{-\alpha_{A}d_{A}}\left(1+T_{23}R_{34}T_{32}e^{-2\alpha_{p}d_{p}}\right).$$

Hence:(75)$$R_{SERS}=T_{12}^{ex}T_{21}e^{-\alpha_{A}^{ex}d_{A}}\left(1+R_{23}^{ex}+T_{23}^{ex}R_{34}^{ex}T_{32}^{ex}e^{-2\alpha_{p}^{ex}d_{p}}\right)e^{-\alpha_{A}d_{A}}\left(1+T_{23}R_{34}T_{32}e^{-2\alpha_{p}d_{p}}\right).$$

The unattenuated SERS spectrum is provided by Equation (42). Typically, the absorption caused by analytes like viruses or bacteria is minimal, especially when using near-infrared excitation, i.e., $\alpha_{A}^{ex}=\alpha_{A}=0$. Thus, Equation (75) simplifies to:(76)$$R_{SERS}=T_{12}^{ex}T_{21}\left(1+R_{23}^{ex}+T_{23}^{ex}R_{34}^{ex}T_{32}^{ex}e^{-2\alpha_{p}^{ex}d_{p}}\right)\left(1+T_{23}R_{34}T_{32}e^{-2\alpha_{p}d_{p}}\right).$$

Equation (76) shows that the shape of the SERS spectrum will be modulated via the optical property of the hot spot layer.

## 5. The Effect of the Optical Attenuation on SERS Quantification

Quantifying SERS involves establishing a quantitative link between the measured SERS peak intensity, ${I^{\prime }}_{SERS}\left(\Delta v\right),$ and the analyte concentration, $n_{A}$, and is very important for SERS-based sensing applications.

When the size of the analytes is significantly smaller than the size of the hot spots in solution-based measurements, quantification can be discussed using Equations (47) and (48). However, several fundamental assumptions need to be made beforehand: (1) in solution-based SERS measurements, all measurements occur at a point where the interaction between PCNs and analytes reaches equilibrium; (2) the concentrations of both PCNs and analytes remain uniform throughout the measurements; and (3) any interfering spectral features, such as baseline signals or background medium, have been removed from the measured SERS spectrum.

In Equation (47), two parameters are related to $n_{A}$: $M_{A}$, the number of analytes adsorbed on each PCN, and $\alpha_{l}$, the attenuation due to optical absorption of the analyte system. The density, $n_{A}^{a}$, of the adsorbed analyte molecules is given by $n_{A}^{a}=n_{PCN}M_{A}$, and $\alpha_{l}$ can be written as:(77)$$\alpha_{l}=\alpha \prime_{LSPR}+\alpha_{bf}+\varepsilon_{A}\;\left(n_{A}-n_{A}^{a}\right),$$

$\alpha \prime_{LSPR}$ is the modified absorption of PCNs. When analyte molecules are adsorbed onto a PCN, the PCN–analyte combination can be considered as a coated particle. Considering the spherical nature of PCNs, where their radius, *r*, is much smaller than $\lambda_{ex}$, according to Ref. [32], $\alpha \prime_{LSPR}$ can be written as follows:(78)$$\alpha \prime_{LSPR}=4\pi r^{3}n_{PCN}Im\left[\frac{\varepsilon_{p}-\varepsilon_{m}}{\varepsilon_{p}+2\varepsilon_{m}}-\frac{a}{r}\frac{\left(\varepsilon_{p}-\varepsilon \prime_{A}\right)\left(\varepsilon_{m}+2\varepsilon \prime_{A}\right)}{\varepsilon_{p}+2\varepsilon_{m}}\right],$$
where *a* represents the diameter of an analyte, with a≪r. $\varepsilon_{m}$ is the dielectric function of the measurement medium, usually $\varepsilon_{m}=1$ (in air) or 78.4 (in water), and $\varepsilon \prime_{A}$ is the effective dielectric function of the analyte coating layer on the PCN particle. The first term in Equation (78), $n_{PCN}Im\left[4\pi r^{3}\frac{\varepsilon_{p}-\varepsilon_{m}}{\varepsilon_{p}+2\varepsilon_{m}}\right]=\alpha_{LSPR}.$ The term $\varepsilon \prime_{A}$ results from a $M_{A}$ analyte coating on a PCN with a layer thickness of *a*, leading to a volume fraction $\delta_{A}=\frac{M_{A}}{24}\frac{a^{2}}{r^{2}}$. According to Equation (59), $\varepsilon \prime_{A}$ can be expressed as:(79)$$\varepsilon \prime_{A}\approx \left(1-\delta_{A}\right)\varepsilon_{m}+\delta_{A}\varepsilon_{A}=\varepsilon_{m}-\frac{M_{A}}{24}\frac{a^{2}}{r^{2}}\varepsilon_{m}+\frac{M_{A}}{24}\frac{a^{2}}{r^{2}}\varepsilon_{A}.$$

Thus, the second term in Equation (78) becomes:(80)$$4\pi r^{3}n_{PCN}Im\left[-\frac{a}{r}\frac{\left(\varepsilon_{p}-\varepsilon \prime_{A}\right)\left(\varepsilon_{m}+2\varepsilon \prime_{A}\right)}{\varepsilon_{p}+2\varepsilon_{m}}\right]\approx -4\pi r^{3}n_{PCN}Im\left(3\varepsilon_{m}\frac{a}{r}\frac{\varepsilon_{p}-\varepsilon_{m}}{\varepsilon_{p}+2\varepsilon_{m}}\right)=-\frac{3\varepsilon_{m}a}{r}\alpha_{LSPR},$$
with $\alpha \prime_{LSPR}=\left(1-\frac{3\varepsilon_{m}a}{r}\right)\alpha_{LSPR}$, which is independent of $M_{A}$. Therefore, according to Equations (47) and (77), the quantitative relationship between $I\prime_{SERS}$ and $n_{A}$ depends on how $n_{A}^{a}$ (or $M_{A}$) correlates with $n_{A}$, which is dominated by the analyte adsorption isotherm on a single PCN particle. Given that both $\alpha \prime_{LSPR}$ and $\alpha_{bf}$ are independent of $n_{A}$, and letting $\alpha_{0}=\alpha \prime_{LSPR}+\alpha_{bf}$, two distinct scenarios emerge from Equation (77). First, if $\alpha_{0}\gg \varepsilon_{A}\;\left(n_{A}-n_{A}^{a}\right)$, Equation (47) can be written as:(81)$$I\prime_{SERS}\left(\Delta v\right)\propto M_{A},$$
i.e., the ${I^{\prime }}_{SERS}\left(\Delta v\right)-n_{A}$ relationship is solely determined by the $M_{A}-n_{A}$ relation, i.e., the analyte adsorption isotherm on a single PCN.

However, if $\alpha_{0}\approx \varepsilon_{A}\;\left(n_{A}-n_{A}^{a}\right)$, or even when $\alpha_{0}<\varepsilon_{A}\;\left(n_{A}-n_{A}^{a}\right)$, i.e., the analyte molecule/particle is highly absorptive in the Raman wavenumber region, the ${I^{\prime }}_{SERS}\left(\Delta v\right)-n_{A}$ relationship becomes quite complicated. Assuming $\varepsilon_{A}\;\left(n_{A}-n_{A}^{a}\right)f\ll 1$, then:(82)$${I^{\prime }}_{SERS}\left(\Delta v\right)\propto n_{A}^{a}\left[1-\varepsilon_{A}\;\left(n_{A}-n_{A}^{a}\right)f\right]e^{-\alpha_{0}f}.$$

Let us examine two well-known adsorption isotherms for Equations (81) and (82): the Langmuir and Freundlich isotherms [34].

For the Langmuir isotherm:(83)$$M_{A}=M_{A}^{0}\Theta_{A}=M_{A}^{0}\frac{n_{A}}{n_{A}+K^{-1}},$$
where $M_{A}^{0}$ is the maximum number of analytes that can be adsorbed on a PCN particle, a constant; $\Theta_{A}$ is the coverage of analytes adsorbed on a PCN particle, and K is the Langmuir equilibrium constant. The black curve in Figure 16A plots the ${I^{\prime }}_{SERS}\left(\Delta v\right)-n_{A}$ based on Equation (81). Clearly the ${I^{\prime }}_{SERS}\left(\Delta v\right)-n_{A}$ exactly follows the Langmuir isotherm trend, with ${I^{\prime }}_{SERS}\left(\Delta v\right)$ monotonically increasing with $n_{A}$, approaching a saturation value. However, when the optical absorption of the solution cannot be neglected, especially at high $n_{A},$ the ${I^{\prime }}_{SERS}\left(\Delta v\right)-n_{A}$ relationship changes significantly, as shown by other colored curves in Figure 16A: ${I^{\prime }}_{SERS}\left(\Delta v\right)$ initially increases monotonically with $n_{A}$; after reaching a critical concentration, ${I^{\prime }}_{SERS}\left(\Delta v\right)$ decreases monotonically with $n_{A}$. This decrease becomes more pronounced, especially at high $n_{A}$, when $\varepsilon_{A}$ is substantial. This phenomenon has been experimentally observed in many SERS measurements [35,36].

For the Freundlich isotherm [34]:(84)$$n_{A}^{a}=n_{PCN}M_{A}=kn_{A}^{1/n},$$
where k and n are constants that determine the Freundlich isotherm. The log–log plot of the black curve in Figure 16B represents the ${I^{\prime }}_{SERS}\left(\Delta v\right)-n_{A}$ relationship based on Equation (81), indicating that ${I^{\prime }}_{SERS}\left(\Delta v\right)-n_{A}$ follows a power law relation, with ${I^{\prime }}_{SERS}\left(\Delta v\right)$ monotonically increasing with $n_{A}$. However, when the solution’s absorption cannot be neglected, according to Equation (82), the ${I^{\prime }}_{SERS}\left(\Delta v\right)-n_{A}$ relationship changes significantly: at low $n_{A},$ the ${I^{\prime }}_{SERS}\left(\Delta v\right)-n_{A}$ still follows a power law, while at high $n_{A}$, ${I^{\prime }}_{SERS}\left(\Delta v\right)$ decreases with $n_{A}$, as shown by other colored curves in Figure 16B.

For cases where the size of the analyte particle significantly exceeds that of the PCN, Equations (31) and (76) show that:(85)$$I\prime_{SERS}^{sphere}\left(\Delta v\right)\propto M_{PCN}n_{A}e^{-a_{l}f},$$
where $M_{PCN}$ is determined by the isotherm depicting how the PCNs adsorb on large particles:(86)$$\alpha_{l}=\alpha_{0}+\varepsilon \prime_{A}n_{A}+\varepsilon_{PCN}\left(n_{PCN}-M_{PCN}n_{A}\right),$$
where $\varepsilon \prime_{A}$ is the effective absorptivity of the hybrid PCN–analyte particle suspension illustrated in Figure 5 and Figure 6, and the last term in Equation (86) accounts for the contribution of LSPR-induced absorption due to freely suspended PCNs. As $n_{A}$ increases, $M_{PCN}$ is expected to decrease. Assuming $n_{PCN}\gg$ $n_{A}$, we can neglect the third term in Equation (86), simplifying $\alpha_{l}$ to $\alpha_{l}\approx \alpha_{0}+\varepsilon \prime_{A}n_{A}$. Regarding $M_{PCN}$, there are limited studies on this scenario, with most results suggesting that this isotherm follows a Langmuir-like pattern as $n_{A}$ increases with a fixed $n_{PCN}$:(87)$$M_{PCN}=M_{PCN}^{0}\frac{n_{A}n_{PCN}}{n_{A}n_{PCN}+K^{-1}},$$
where $M_{PCN}^{0}$ denotes the saturation number of PCN particles absorbed on an analyte particle. Therefore, for a fixed $n_{PCN}$, Equation (85) is changed to:(88)$$I\prime_{SERS}^{sphere}\left(\Delta v\right)\propto \frac{Kn_{A}}{K+n_{A}}e^{-\left[\alpha_{0}+\varepsilon \prime_{A}n_{A}\right]f}.$$

This leads to a similar quantitative relationship, as shown in Figure 16A. However, it could also be argued that since $n_{PCN}\gg$ $n_{A}$, at low $n_{A}$, the number of PCN particles on each analyte particle already reaches saturation. Consequently, Equation (88) can be further simplified as:(89)$$I\prime_{SERS}^{sphere}\left(\Delta v\right)\propto n_{A}M_{PCN}^{0}e^{-\left[\alpha_{0}+\varepsilon_{A}n_{A}\right]f}.$$

Figure 16C plots how ${I^{\prime }}_{SERS}\left(\Delta v\right)-n_{A}$ changes based on Equation (89): In the absence of attenuation effects, ${I^{\prime }}_{SERS}\left(\Delta v\right)-n_{A}$ follows a linear relationship. When $\varepsilon \prime_{A}$ starts to influence the system, particularly at high $n_{A}$, ${I^{\prime }}_{SERS}\left(\Delta v\right)$ deviates from this linear pattern. The extent of this deviation increases with higher $\varepsilon \prime_{A}$. When $\varepsilon \prime_{A}$ becomes sufficiently large, ${I^{\prime }}_{SERS}\left(\Delta v\right)$ b starts to decrease as $n_{A}$ increases.

These findings emphasize that the quantification of solution-based SERS measurements not only depends on the analyte–PCN adsorption isotherm but also significantly on the optical properties of the analyte–PCN system. If the analyte–PCN system exhibits substantial optical absorption within the SERS measurement wavenumber range, the quantification will be profoundly impacted by the optical absorption characteristics of the measurement system.

Similar reasoning can be applied to thin-film-based SERS substrates by exploring Equations (62), (68), and (73).

## 6. The Effect of the Optical Attenuation on Florescence Background

According to Equation (1), the real collected SERS spectrum includes featureless fluorescence signals or other scattered signals originating from the SERS substrates, contributing to the overall baselines in the spectra [37,38,39]. These baseline signals also stem from the measured volume in various substrate configurations and undergo similar optical attenuation, as discussed in Section 4. Consequently, contingent upon the localized variations in the optical properties of the SERS substrate, the amplitude and shape of the baseline can undergo significant changes.

To exemplify the attenuation effect on the baseline, we artificially introduced an exponential decay baseline for the BPE spectrum, depicted as the red spectra in Figure 13B. The resulting spectrum, based on the four absorption curves for $\alpha_{l}$ shown in Figure 13A, is plotted as the blue spectra in Figure 17. With varied spectra in $\alpha_{l}$, optical attenuation not only changes the spectral shape as the relative peak intensities vary but also modifies the baseline shape: For Cases 1 and 2 shown in Figure 17A,B, although both baselines exhibit a monotonic decrease with Δv, the amplitude and shape of the two baselines differ significantly. Case 1 only induces a slight modulation in the spectrum and the baseline, whereas Case 2 substantially decreases the overall intensity of the SERS spectrum and reduces the baseline amplitude. In Cases 3 and 4, the baselines no longer follow a monotonic pattern with Δv. Instead, both baselines resemble a parabolic shape with uneven attenuation in the small and large Δv regions.

## 7. Conclusions

In summary, within this comprehensive theoretical framework, the intricate dynamics affecting SERS measurements in both solution and thin-film configurations have been systematically analyzed. This analysis takes into account the specific SERS substrates utilized and the dimensions of the target analytes, elucidating the complex interplay of various factors.

When the analytes are much smaller than the hot spot size, the effective SERS EF is intricately influenced by factors like the quantity of analytes adsorbed on hot spot sites, the dimensions (volumes) of the hot spots, the orientation of analytes on these sites, and the polarization of the excitation laser. These variables collectively impact both the intensity and shape of the measured SERS spectrum. Notably, different SERS peaks corresponding to the same analyte may not possess identical EFs on the same substrate or even at different concentrations due to these multifaceted factors. In the case of analytes significantly larger than the hot spots, only open hot spots accessible to the analyte contribute to the SERS signal. This scenario presents a challenge in defining a specific SERS EF. Therefore, considering the entire SERS spectrum provides a more realistic representation, the shape of the spectrum depends on the distance-dependent local electric field and the heterogeneity of the analyte particle.

By carefully examining the paths of excitation laser propagation and the back-collected SERS signal, it becomes evident that the optical properties of the substrate–analyte system play pivotal roles in reshaping the SERS spectrum. Through rigorous analysis, it has been demonstrated that accounting for the optical properties of SERS substrates allows for the uneven tuning of relative SERS intensity at different wavenumbers, leading to spectral distortion. This effect is particularly pronounced when $\lambda_{ex}$ approximates $\lambda_{LSPR}$, indicating that the optical characteristics of PCNs or thin films can significantly alter the resulting spectrum. By incorporating the effective medium theory into the derivations, explicit relationships between SERS intensity and analyte concentration can be established. These results demonstrate that the optical attenuation due to the optical properties of the SERS substrate–analyte system profoundly influences SERS quantification, introducing significant variations in SERS baselines during measurements. However, establishing a direct correlation between the optical absorption spectrum and SERS enhancement poses experimental challenges due to several factors: First, the optical absorption of SERS substrates is highly localized in nature, and most of the time it remains unknown. Second, the distribution of the morphology of nanostructures leads to a varied distribution of hot spots with different sizes and $\lambda_{LSPR}$, introducing additional variables. Third, the simultaneous measurement of both SERS and UV–Vis at identical locations on SERS substrates is complicated.

Nevertheless, this theoretical framework provides profound insights into observed phenomena in day-to-day measurements, emphasizing the localization nature of SERS. It reveals that different locations on the same substrate, even with identical analytes, display diverse local optical properties, leading to significant spectral variations. The outcomes derived from this theory can be instrumental in comprehending and interpreting measured SERS spectra across various analyte–SERS substrate setups. Moreover, these findings can serve as a guiding principle for designing SERS substrates and optimizing SERS instrument configurations.

## Figures and Tables

**Figure 1 nanomaterials-13-02998-f001:**
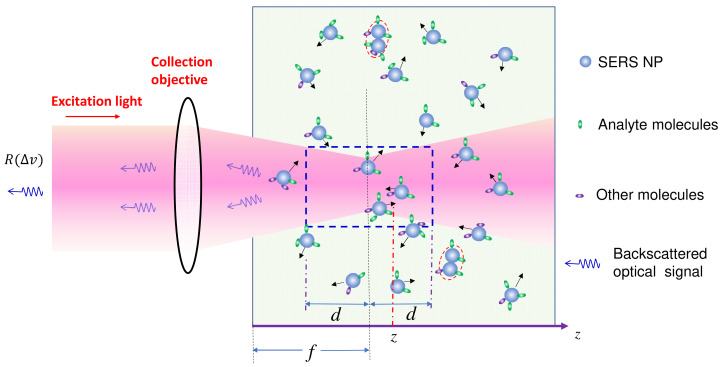
The schematics of the solution-based SERS measurement. The black arrows denote the Brownian motion direction of each PCN.

**Figure 2 nanomaterials-13-02998-f002:**
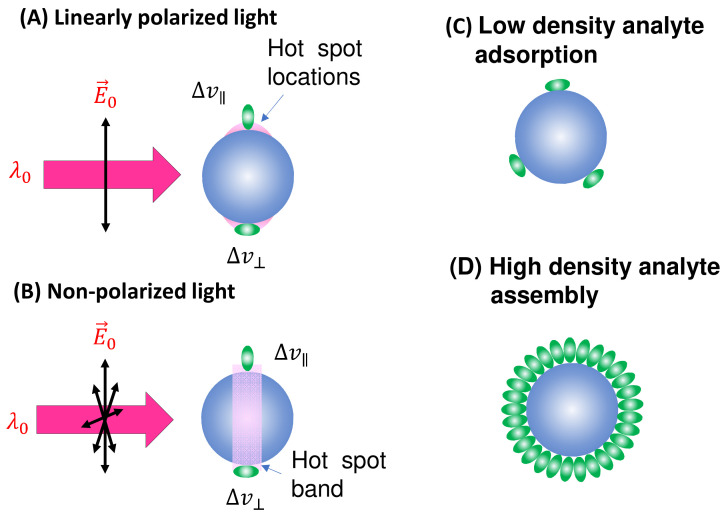
The spherical PCN for SERS measurement: (**A**) linearly polarized and (**B**) non-polarized excitation and possible analyte molecule orientation on a PCN. The pink shaded areas depict the locations of hot spots. The configuration of (**C**) low-coverage and (**D**) high-coverage analyte adsorption on a PCN.

**Figure 4 nanomaterials-13-02998-f004:**
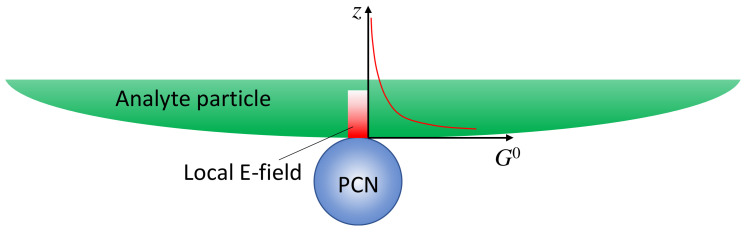
The hot spot distance effect when a PCN is adsorbed on a large analyte particle.

**Figure 5 nanomaterials-13-02998-f005:**
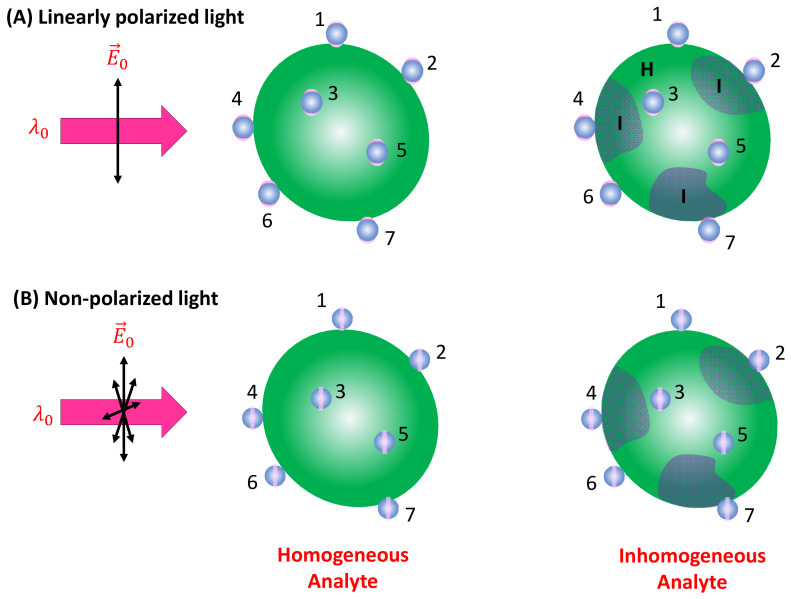
(**A**) The linearly polarized and (**B**) non-polarized excitation of PCNs adsorbed on a large homogenous and inhomogeneous analyte particle. The numbers in the figure indicate different PCNs (or locations) on the analyte particle.

**Figure 6 nanomaterials-13-02998-f006:**
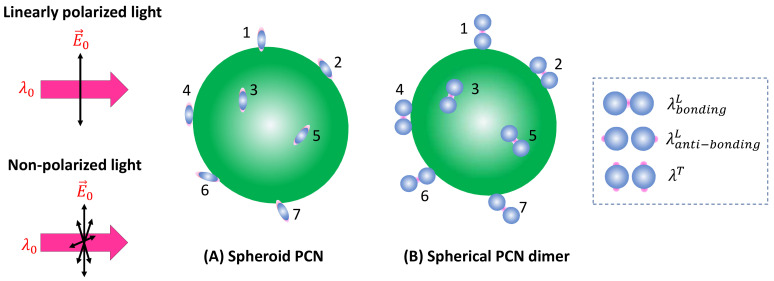
The adsorption configuration of (**A**) spheroid PCNs and (**B**) spherical PCN dimers on a large analyte particle. The numbers in the figure indicate different PCNs (or locations) on the analyte particle.

**Figure 7 nanomaterials-13-02998-f007:**
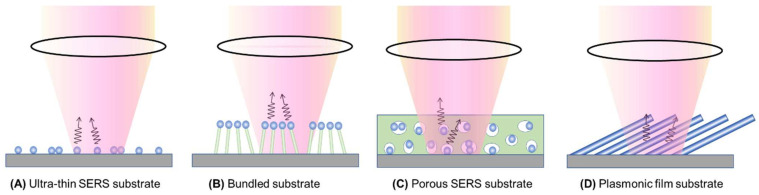
Typical film-based SERS substrates: (**A**) ultra-thin SERS substrate; (**B**) bundled SERS substrate; (**C**) porous SERS substrate; and (**D**) plasmonic film substrate.

**Figure 8 nanomaterials-13-02998-f008:**
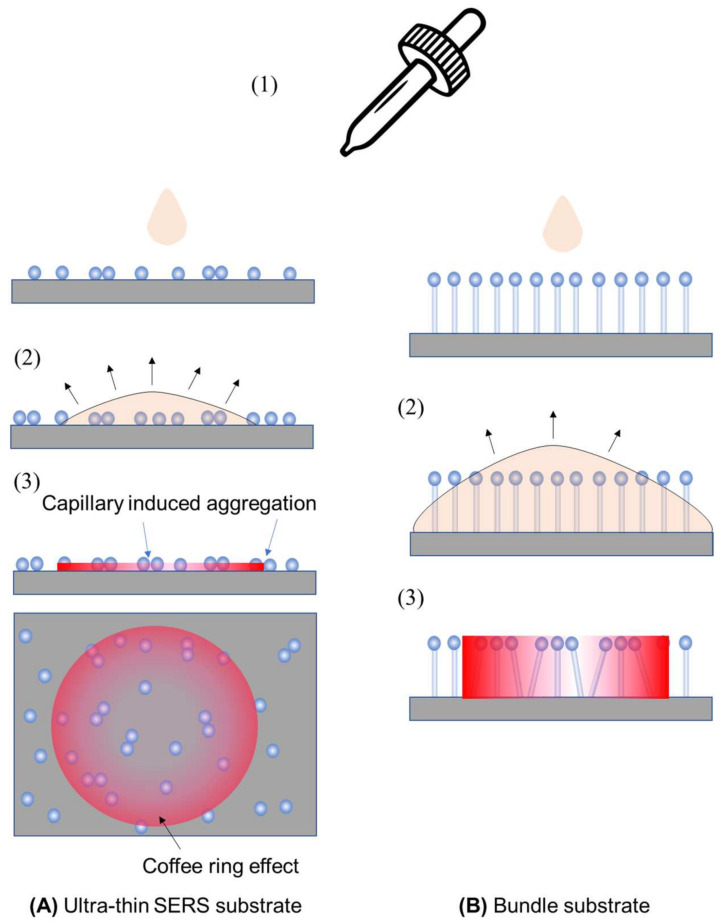
The drop-casting sample preparation method for (**A**) ultra-thin SERS substrates and (**B**) bundle substrates. Step 1: dispensing the analyte droplet; step 2: droplet spreading; and step 3: spatial distribution of analyte concentration on substrates after dewetting.

**Figure 9 nanomaterials-13-02998-f009:**
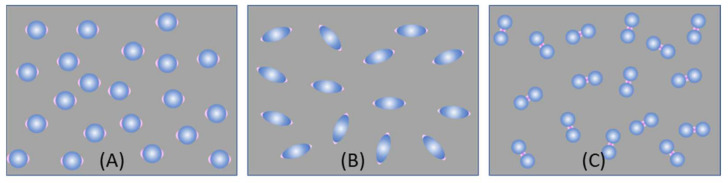
Three possible ultra-thin SERS substrates: (**A**) spherical PCNs; (**B**) spheroid PCNs; and (**C**) spherical PCN dimers.

**Figure 10 nanomaterials-13-02998-f010:**
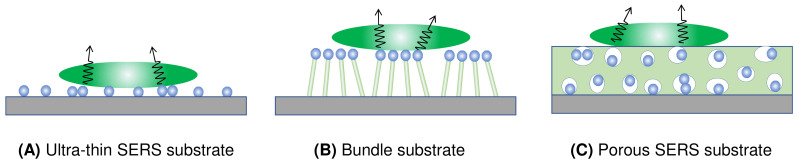
Large analyte particle on an (**A**) ultra-thin SERS substrate; (**B**) bundled or plasmonic SERS substrate; and (**C**) porous SERS substrate.

**Figure 11 nanomaterials-13-02998-f011:**
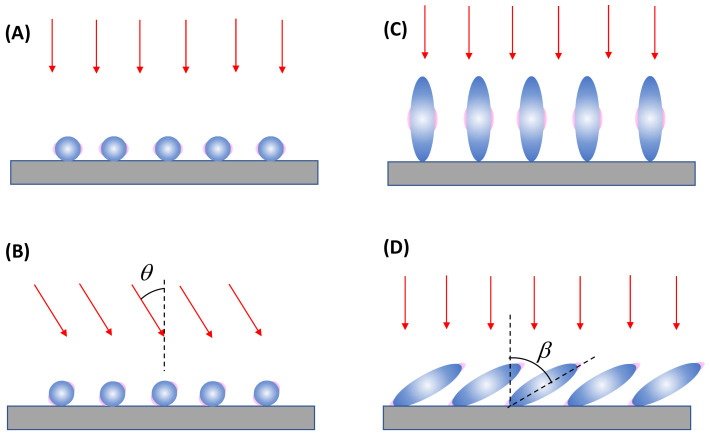
Illustration of potential hot spot locations for two thin-film-based SERS substrates under different optical illuminations and structure configurations: (**A**) Normal incident and (**B**) tilted excitation on a spherical PCN thin film. Normal excitation on the (**C**) vertically and (**D**) tilted aligned spheroid PCN thin film.

**Figure 12 nanomaterials-13-02998-f012:**
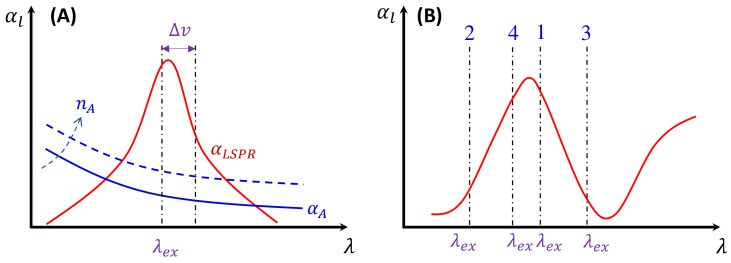
(**A**) The illustration of sources for the absorption of an analyte–PCN solution. The solid blue curve is due to absorption $\alpha_{A}$ of an analyte solution, the dashed blue curve shows the changed $\alpha_{A}$ at an increased analyte concentration $n_{A}$, and the red curve is due to the LSPR of PCNs. (**B**) The illustration of the excitation wavelength in different regions (labeled 1–4) of the absorption curve.

**Figure 13 nanomaterials-13-02998-f013:**
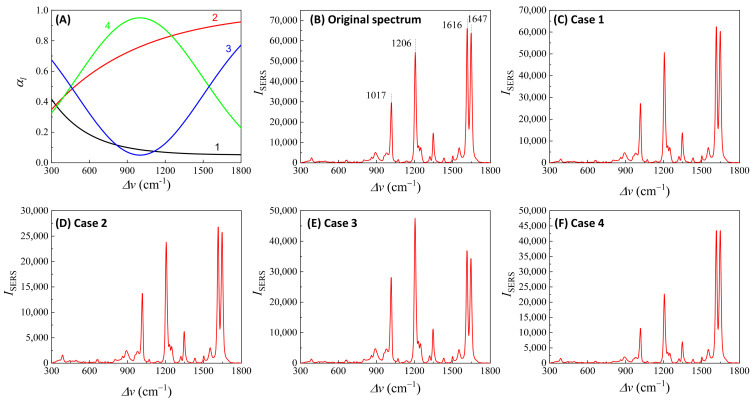
(**A**) The absorption spectra in the SERS measurement region according to excitation wavelengths marked 1–4 in Figure 12B. (**B**) The experimental SERS spectrum of BPE. (**C**–**F**) The calculated distorted SERS spectra based on absorption spectra 1–4 in (**A**) based on Equation (47).

**Figure 14 nanomaterials-13-02998-f014:**
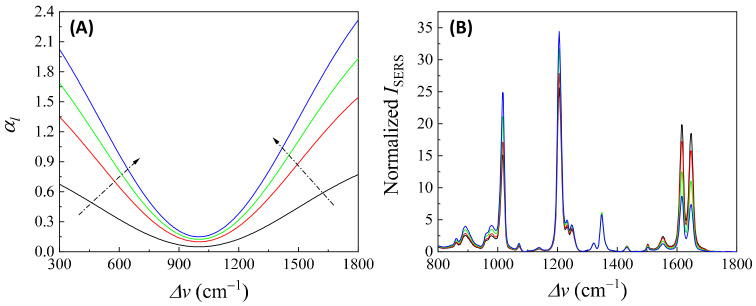
(**A**) The $n_{A}$-dependent $\alpha_{l}\left(\Delta v\right)$. The arrows show the increase in $n_{A}$. (**B**) The area normalized distorted SERS spectra due to different $\alpha_{l}\left(\Delta v\right)$ in (**A**) based on Equation (47) and $\alpha_{l}\propto \;n_{A}$.

**Figure 15 nanomaterials-13-02998-f015:**
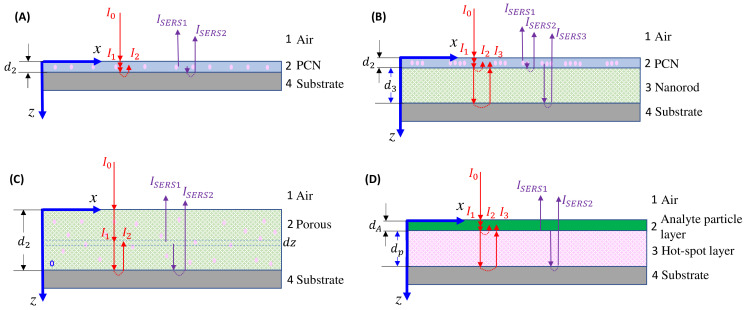
Excitation laser (red) and scattered light (purple) propagation paths in (**A**) ultra-thin-film; (**B**) bundled thin-film; and (**C**) porous thin-film SERS substrates for small analytes. (**D**) Large analyte on a thin-film SERS substrate.

**Figure 16 nanomaterials-13-02998-f016:**
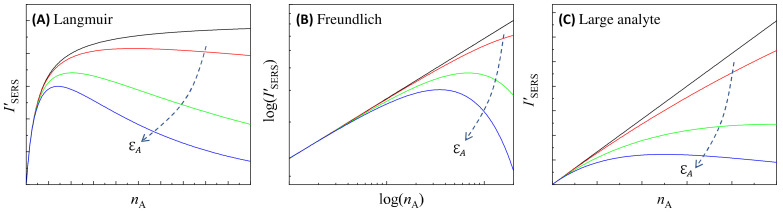
The illustration of the ${I^{\prime }}_{SERS}\left(\Delta v\right)-n_{A}$ relationship of different $\varepsilon_{A}$ for (**A**) the Langmuir isotherm, (**B**) the Freundlich isotherm, and (**C**) large analyte particles, calculated based on Equations (81) and (82). The dashed arrows in plots show the increase in $\varepsilon_{A}$. The colored curves, from black, to red, green, and blue, show the quantification relationship with increase in $\varepsilon_{A}$.

**Figure 17 nanomaterials-13-02998-f017:**
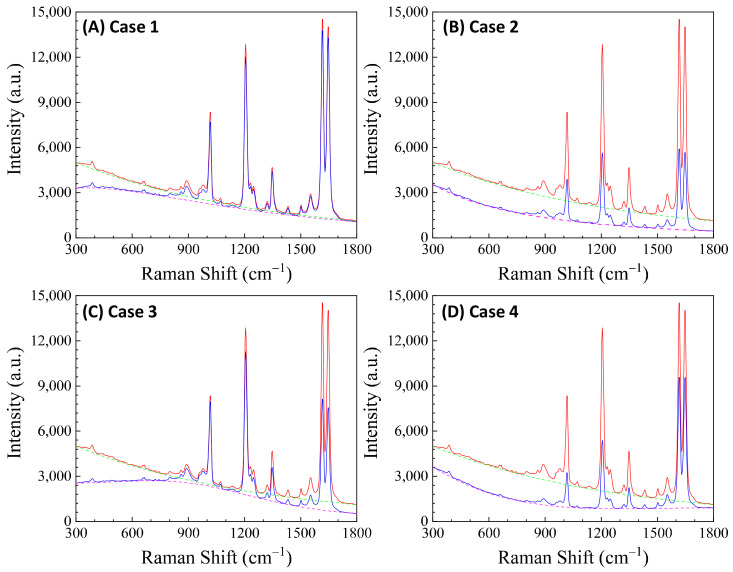
The optical attenuation effect on the SERS baseline: the red curves are the original spectrum with an exponential decay baseline, and the blue curves are the distorted SERS spectra based on absorption spectra 1–4 (corresponding to (**A**–**D**)) in Figure 13A calculated based on Equation (47). All dashed curves highlight baselines for the corresponding SERS spectra.

## Data Availability

All relevant data are within the paper.

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
