# Peer review of "On the Measurements of the Surface-Enhanced Raman Scattering Spectrum: Effective Enhancement Factor, Optical Configuration, Spectral Distortion, and Baseline Variation"

_nanomaterials, 2023, doi:10.3390/nano13232998_

Round 1
Reviewer 1 Report
Comments and Suggestions for Authors
This manuscript reports on comparative analysis of various parameters related with plasmonic nanoparticles and nanostructured substrates as well as the properties of adsorbed analyte molecules on the SERS enhancement factor and shape of the spectra. The analysis was supported by presentation of specific and clear figures. Corrections to obtain more reliable enhancement factor for various SERS active structures are provided. This manuscript provides some insights into the SERS measurements technology. The manuscript is clearly written. I recommend publication in NANOMATERIALS after considering the several points provided below:
1) The SERS enhancement factor critically depends on the position of excitation wavelength with respect to the plasmon resonance absorption. For well-defined nanostructures it was demonstrated that maximum SERS EF can be obtained at excitation wavelength slightly blue shifted with respect to plasmon resonance peak. In this case both incident and scattered photons are enhanced. This matter should be discussed in the manuscript.
2) In many cases direct correlation between the UV-Vis extinction maximum and SERS enhancement is not possible, because of absorption at hot spots is unknown. This matter should be discussed in the manuscript.
Reviewer 2 Report
Comments and Suggestions for Authors
In this review, the author explains the theoretical effect behind the enhancement factor onto different types of SERS substrates. Although much work has been done by the author, for considering the accept of this paper, the following issues need to be solved.
1. In general, the manuscript has a lack of references. Considering this article as a review 36 references are not enough. The author should introduce references in the general introduction and in the introduction of each chapter and subchapter.
2. In general, the references have more than 10 years, the author should update the references in all chapters.
3. The author should reference some scientific articles when they explain the behaviour observed in each type of SERS substrate.
4. The author should take care with the format of the references, he is using different layouts.
5. In line 66, the author claims ¨plasmonic colloidal particles (PCNs) are uniformly dispersed in the analyte solution¨ and they based all the theoretical explanations in this uniformity, nevertheless, if we have aggregates what would happen?
6. In general, some equations showed in the article have been developed before by other Scientifics, the author should indicate the proper reference in this equations.
7. Chapter 3.1.1 and 3.2.1 have the same title, the same with 3.1.2 and 3.2.2, I recommend to rename them.
8. In line 546, reference 19 is not an example of ¨porous inorganic or organic films as host for plasmonic nanoparticles dispersed into the pores¨.
9. In the case of ¨the analyte molecules are much larger than the size of hot-spots¨ the author should comment the inhomogeneity of the signal due to the different parts of the ¨molecule¨ (virus, bacteria…) exposed to the light.
10. The results of Figure 12, Figure 13, Figure 14, Figure 16, Figure 17 have been measured by the author or they have been extracted from the literature? In the first case, the author should indicate the experimental details (Raman equipment, type of SERS substrate and composition, BPE concentration…) and in the second case they should indicate the original article.
Round 2
Reviewer 2 Report
Comments and Suggestions for Authors
I think that the author has made appropriate additional changes to the manuscript, which is acceptable for publication.